# A selective gut bacterial bile salt hydrolase alters host metabolism

Lina Yao[1], Sarah Craven Seaton[1†], Sula Ndousse-Fetter[1], Arijit A Adhikari[1], Nicholas DiBenedetto[2], Amir I Mina[3‡], Alexander S Banks[3], Lynn Bry[2], A Sloan Devlin[1]*

[1]Department of Biological Chemistry and Molecular Pharmacology, Harvard Medical School, Boston, United States; [2]Department of Pathology, Massachusetts Host-Microbiome Center, Brigham and Women's Hospital, Boston, United States; [3]Division of Endocrinology, Diabetes and Hypertension, Brigham and Women's Hospital, Boston, United States

**Abstract** The human gut microbiota impacts host metabolism and has been implicated in the pathophysiology of obesity and metabolic syndromes. However, defining the roles of specific microbial activities and metabolites on host phenotypes has proven challenging due to the complexity of the microbiome-host ecosystem. Here, we identify strains from the abundant gut bacterial phylum Bacteroidetes that display selective bile salt hydrolase (BSH) activity. Using isogenic strains of wild-type and BSH-deleted *Bacteroides thetaiotaomicron*, we selectively modulated the levels of the bile acid tauro-β-muricholic acid in monocolonized gnotobiotic mice. *B. thetaiotaomicron* BSH mutant-colonized mice displayed altered metabolism, including reduced weight gain and respiratory exchange ratios, as well as transcriptional changes in metabolic, circadian rhythm, and immune pathways in the gut and liver. Our results demonstrate that metabolites generated by a single microbial gene and enzymatic activity can profoundly alter host metabolism and gene expression at local and organism-level scales.
DOI: https://doi.org/10.7554/eLife.37182.001

*For correspondence:
sloan_devlin@hms.harvard.edu

Present address: †Indigo Agriculture, Boston, United States; ‡University of Pittsburgh School of Medicine, Pittsburgh, United States

## Introduction

The human gut microbiome is known to play a crucial role in human energy harvest and homeostasis (*BäckhedBackhed et al., 2004*; *Turnbaugh et al., 2006*). Lean and obese people harbor different gut bacterial communities, suggesting that developing gut bacterial imbalances may contribute to obesity (*Ley et al., 2006*; *Turnbaugh et al., 2006*; *Turnbaugh et al., 2008*). Importantly, transplantation of the fecal microbiota from obese humans to germ-free (GF) mice has been shown to result in the development of obesity-associated metabolic phenotypes in recipient mice (*Ridaura et al., 2013*). These studies establish a causal relationship between gut bacteria and host metabolic status. The molecular mechanisms by which gut microbes regulate host metabolism, however, remain largely unknown. This lack of mechanistic understanding regarding the functions of microbial species and their metabolic capabilities has limited the effectiveness of both dietary and therapeutic approaches to improving host physiology (*Jia et al., 2008*; *Wallace et al., 2010*).

The investigation of microbial metabolite production represents both an important opportunity and a challenge in the search to uncover the causal underpinnings of the effects of gut bacteria on host metabolism. One of the most concrete effects that human-associated bacteria have on the host is the production of small molecule metabolites, some of which accumulate to levels in the body higher than that of a typical drug (*Donia and Fischbach, 2015*). Recent research suggests that bacterial metabolites play important roles in host metabolism by regulating host glucose and energy homeostasis (*De Vadder et al., 2014*; *Gao et al., 2009*; *Todesco et al., 1991*). The complexity of

**eLife digest** The microbiome, the collection of bacteria that live in and on human bodies, has a strong influence on how well the body works. However, the diversity of the microbiome makes it difficult to untangle exactly how it has these effects. For example, it is poorly understood how the hundreds of species of bacteria that live in the gut affect metabolism – the chemical processes that make life possible. But they are known to influence how metabolic diseases like diabetes and obesity develop.

When we eat a meal, the body releases compounds called bile acids to help to digest the food. Once the bile acids reach the colon, the bacteria residing there use enzymes to chemically modify the compounds. Imbalances in the resulting pool of over 50 different bile acids may accelerate how quickly people develop metabolic disorders. It is not clear, however, which bile acids have helpful or harmful effects on metabolism.

Yao et al. first identified a selective version of a prevalent gut bacterial enzyme called a bile salt hydrolase. This enzyme was then deleted from a common gut bacterium using genetic tools. Finally, Yao et al. colonized mice lacking any bacteria (i.e., germ-free mice) with either the original bacterium or the hydrolase-deleted bacterium. Mice colonized with the hydrolase-deleted bacteria gained less weight on a high fat diet and had lower levels of fat in their blood and liver. These mice also shifted to burning fats instead of carbohydrates for energy.

The changes in the bile acid pool produced in mice colonized with hydrolase-deleted bacteria did not only affect metabolism. Yao et al. found differences in the activity of genes important for other biological processes as well, such as those that control circadian rhythms and immune responses.

Further research is needed to investigate whether limiting the activity of the bile salt hydrolase enzyme has similar effects in humans. If so, developing drugs or probiotics that target the enzyme could lead to new treatments for people with metabolic diseases like obesity and fatty liver disease. Investigating the biological effects of other bacterially modified bile acids may identify other possible treatments as well.

DOI: https://doi.org/10.7554/eLife.37182.002

gut microbial ecosystems and associated microbial and host-derived microbial metabolites, however, presents significant obstacles on the path to defining how individual compounds elicit specific in vivo effects. Means to control specific metabolites is critical to understanding how these molecules affect host physiology. In this work, we selectively modulate the in vivo levels of bile acids and demonstrate that this controlled alteration of the metabolite pool exerts distinct effects on host physiology.

Bile acids are steroidal natural products that are synthesized from cholesterol in the liver and constitute an important part of the molecular environment of a healthy human gut (*Ridlon et al., 2006*). Upon ingestion of a meal, bile acids are secreted from the liver and gallbladder into the duodenum where, with the activities of pancreatic enzymes, they form micelles that solubilize lipids and fat-soluble vitamins that are otherwise poorly absorbed. Remaining free bile acids are efficiently reabsorbed from the ileum via the action of bile acid transporters and recirculated back to the liver. Approximately 3–5% of bile acids escape enterohepatic recirculation and enter the colon at a rate of 400–800 mg/day, forming a concentrated pool of metabolites (200 to 1000 µM) (*Hamilton et al., 2007*). In the colon, these molecules are modified by the resident bacteria in near-quantitative fashion, forming a class of on the order of 50 different metabolites called secondary bile acids (*Figure 1A*). In addition to their role in digestion, many primary and secondary bile acids act as ligands for host nuclear receptors, including the farnesoid X receptor (FXR), the pregnane X receptor (PXR), the vitamin D receptor (VDR), the liver X receptor (LXR) and the G-protein-coupled receptor TGR5 (*Fiorucci and Distrutti, 2015*; *Katsuma et al., 2005*; *Makishima et al., 2002*; *Song et al., 2000*; *Staudinger et al., 2001*). By acting as agonists or antagonists for these receptors, bile acids further impact the regulation of glucose tolerance and homeostasis, insulin sensitivity, lipid metabolism, triglyceride and cholesterol levels, and energy expenditure by the host (*Fiorucci and Distrutti, 2015*; *Modica et al., 2010*). Additionally, bile acids regulate their own biosynthesis via an FXR-mediated negative feedback mechanism, which affects downstream nutrient availability for the host

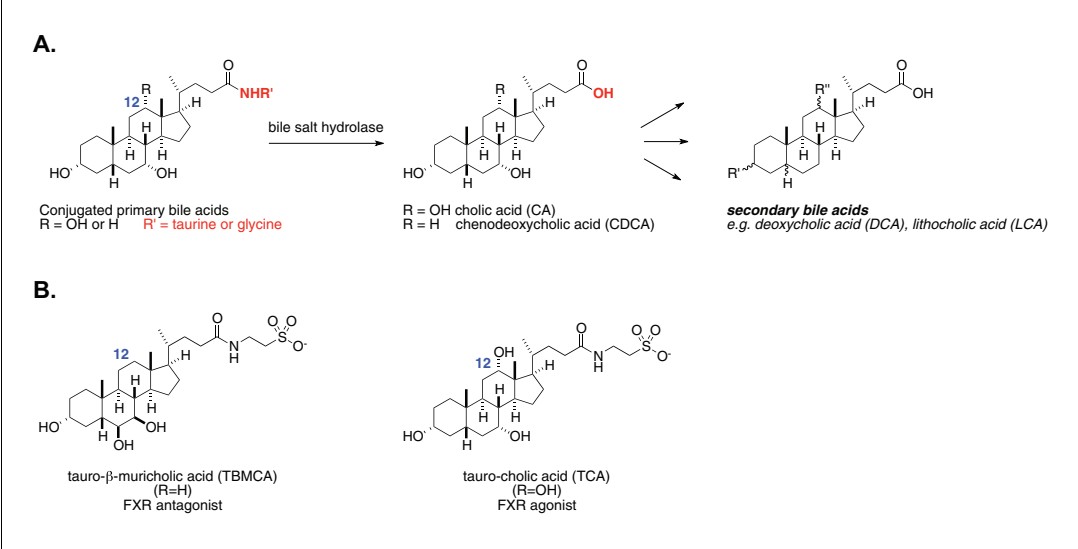

**Figure 1.** Enzymatic activity of gut bacterial bile salt hydrolase (BSH) enzymes. (**A**) BSH cleave the amide bond linking primary bile acids to taurine or glycine. (**B**) Structures of the two most abundant murine-conjugated bile acids, tauro-β-muricholic acid (TβMCA) and tauro-cholic acid (TCA).
DOI: https://doi.org/10.7554/eLife.37182.003

(*Modica et al., 2010*). As a result of these interactions, bile acid imbalance has been implicated as having a causal effect in the development of diet-induced obesity (*Fiorucci and Distrutti, 2015*). Conversely, modification of the bile acid pool by commensal bacteria has been suggested to induce beneficial changes in host metabolism (*Joyce et al., 2014*).

The mechanisms underlying these effects, however, remain largely undefined. Due to the large number of compounds and receptors involved as well as the additional role of bile acids as biological detergents, the in vivo roles of specific bile acids have been difficult to untangle. Our novel approach to deconvoluting the physiological role of structurally distinct bile acids is to control the in vivo activity of selective bacterial bile salt hydrolases (BSH). BSH hydrolyze conjugated bile acids that have been linked to either taurine or glycine by host liver enzymes, revealing unconjugated bile acids (*Figure 1A*). This deconjugation step occurs prior to subsequent bacterial conversion of primary bile acids (e.g. cholic acid and chenodeoxycholic acid) to secondary bile acids (e.g. deoxycholic acid and lithocholic acid) (*Ridlon et al., 2006*).

Prior work suggests that BSH play a critical role in regulating host metabolism. However, these studies have not yet uncovered how specific bile acid metabolites exert their in vivo effects on host metabolism, and conflicting results have been reported regarding whether BSH activity should be increased or decreased to achieve host metabolic benefits (*Joyce et al., 2014*; *Li et al., 2013*). Research efforts to date have either examined correlative relationships between BSH activities, bile acid levels, and metabolic indications (*Li et al., 2013*) or investigated the metabolic effects of 'unconjugated' versus 'conjugated' groups of bile acids (*Joyce et al., 2014*). It is imperative to be able to differentiate bile acids in vivo based on their structure in order to understand their effects on host metabolism. As an important example, taurocholic acid (TCA) and tauro-β-muricholic acid (TβMCA) are both conjugated bile acids but exert different physiological effects: TCA is an FXR agonist, while TβMCA is an FXR antagonist (*Figure 1B*) (*Li et al., 2013*; *Sayin et al., 2013*).

Herein, we uncover a group of bacteria within the abundant human gut commensal genus *Bacteroides* that possess selective BSH activity. We then identify the gene responsible for this activity in *Bacteroides thetaiotaomicron* and construct a knockout strain. By monocolonizing germ-free (GF) mice with the wild-type or BSH-deleted strain, we demonstrate that we can predictably alter the in vivo bile acid pool using this selective enzyme and that this change has significant effects on host metabolic status. Our results demonstrate that the deletion of a single bacterial gene can exert significant effects on host metabolism in a gnotobiotic environment and highlight the importance of modulating specific compounds when seeking to understand the effects of bacterial metabolites on host physiology.

## Results

### Selected species of *Bacteroides* accept distinct bile acid cores as BSH substrates

BSH (EC 3.5.1.24) are found across a wide range of bacterial genera from the two dominant gut phyla, Bacteroidetes and Firmicutes (*Jones et al., 2008*). However, the structural and activity characterization of these enzymes has been largely limited to Gram-positive species (i.e. *Clostridia*, *Lactobacillus*, *Bifidobacterium*, *Listeria*) (*Begley et al., 2006*; *Rossocha et al., 2005*). These enzymes largely demonstrate non-selective activities, cleaving all conjugated bile acids independent of either the bile acid core or amino acid conjugate (taurine or glycine) (*Ridlon et al., 2006*). While differential reactivity toward conjugated substrates has been observed in some Gram-positive strains, in these cases, the selectivity has been based on a preference for one amino acid over the other, not on the structure of the steroidal core (*De Boever P and Verstraete, 1999*; *Grill et al., 1995*; *Kim et al., 2004*; *Ridlon et al., 2006*). In contrast, the activity of Gram-negative bacteria has been largely underexplored. While *Bacteroides fragilis* ATCC 25285 was reported to exhibit non-selective BSH activity (*Stellwag and Hylemon, 1976*), some *Bacteroides vulgatus* strains were observed to cleave taurochenodeoxycholic acid (TCDCA) and TβMCA but minimally cleaved TCA (*Chikai et al., 1987*; *Kawamoto et al., 1989*), thus exhibiting a degree of selectivity based on the hydroxylation pattern of the steroid. These results suggested to us that perhaps other strains within the phylum Bacteroidetes might display steroidal core-based selectivity. To investigate this question, we performed a screen of the BSH activity of twenty Bacteroidetes strains found in the human gut (*Figure 2A* and *Figure 2—figure supplement 1* and *2*) (*Kraal et al., 2014*). We also tested *Clostridium perfringens* and *Lactobacillus plantarum*, two Gram-positive species with known non-selective BSH activities, for comparison. We incubated pre-log phase cultures of individual strains with a group of either the most abundant tauro- or the most abundant glyco-conjugated bile acids found in the human and murine GI tracts. We monitored deconjugation over time by UPLC-MS and determined that all hydrolysis reactions had reached steady state by 48 hr (*Figure 2B*, *Figure 2—figure supplement 3*). We then quenched the cultures and profiled bacterial bile acid metabolism. As expected, *C. perfringens* ATCC 13124, *L. plantarum* WCFS1, and *B. fragilis* ATCC 25285 deconjugated all conjugated bile acid substrates tested. Strikingly, the majority of Bacteroidetes strains tested displayed some degree of selectivity for conjugated bile acid substrates, with a preference for deconjugating tauro- over glyco-conjugated substrates. A subset of these strains (*B. thetaiotaomicron* VPI-5482, *B. caccae* ATCC 43185, *B. fragilis* 638R, *Bacteroides* sp. D2, and *Bacteroides* sp. 2_1_16; Group I – red, *Figure 2A*) exhibited selectivity exclusively based on the steroidal core structure, deconjugating C12 = H primary bile acids (i.e. TCDCA, GCDCA, and TβMCA) but not C12 = OH primary bile acids (i.e. TCA and GCA).

To our knowledge, this study represents the first systematic evaluation of BSH activity in the common gut-bacterial phylum Bacteroidetes. Given that specific conjugated and unconjugated bile acids bind to different host receptors and have the potential to exert different downstream effects, the selectivity uncovered here may have important physiological consequences depending on which *Bacteroides* species colonize the host. To further explore this possibility and define the effects of selective BSH on host physiology, we monocolonized GF mice with isogenic strains of wild-type and BSH-deleted *B. thetaiotaomicron* as described below.

### BT2086 is responsible for BSH activity in *B. thetaiotaomicron*

We recognized that deletion of the BSH enzyme from one of the Group I *Bacteroides* species would provide us with a paired set of isogenic strains (wild-type and knockout) that would allow us to rationally manipulate the in vivo bile acid pool in a highly specific manner. In mice, the two most abundant primary bile acids are TCA and TβMCA (*Sayin et al., 2013*). Based on the observed selectivity for deconjugating C12 = H but not C12 = OH core primary bile acids, we predicted that colonization with a BSH wild-type strain would result in lower levels of TβMCA (C12 = H) relative to knockout colonized mice, while the levels of TCA (C12 = OH) in both groups would remain constant. All the five Group I strains displayed weak-to-moderate deconjugation of TβMCA in vitro (*Figure 2A*). Importantly, we did not detect any products of TCA deconjugation from any of these strains. This result suggested that the levels of deconjugated CA in mice colonized with these bacteria would remain

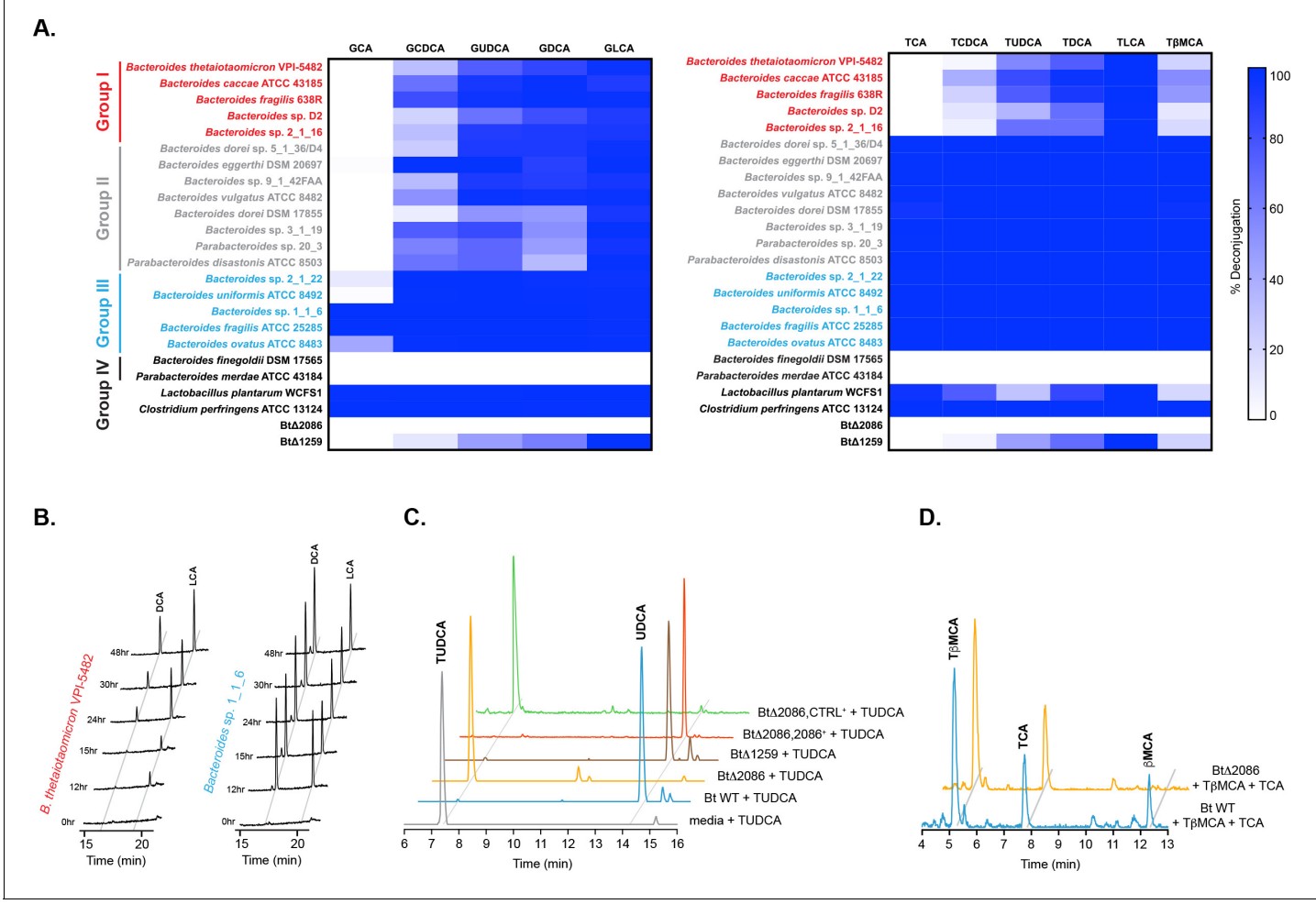

**Figure 2.** Identification of selective BSH activity in the human gut bacterial phylum Bacteroidetes. (**A**) Deconjugation ability of twenty prevalent Bacteroidetes strains and two Firmicutes strains found in the human gut represented as heat maps. Individual strains were incubated for 48 hr total with a group of glyco- or tauro-conjugated bile acids found in human and murine GI tracts. G (glyco-), T (tauro-), CA (cholic acid), CDCA (chenodeoxycholic acid), UDCA (ursodeoxycholic acid), DCA (deoxycholic acid), LCA (lithocholic acid), βMCA (β-muricholic acid). Assays were performed in biological duplicate. Group I (red): Bacteroidetes species that deconjugate primary bile acids based on steroidal core structure (C12 = H but not C12 = OH); Group II (gray): species that deconjugate based on amino acid conjugate; Group III (blue): species that deconjugate all bile acid substrates; Group IV (black): no deconjugation observed. (**B**) Representative UPLC-MS timecourses for deconjugation of TDCA and TLCA showing that steady state has been reached by 48 hr. (**C**) Representative UPLC-MS traces showing that *Bacteroides thetaiotaomicron* wild-type (Bt WT) and BtΔ1259 deconjugate TUDCA, whereas BtΔ2086 does not. BtΔ2086,2086 + recovered the deconjugation function while the BtΔ2086,CTRL +control strain containing an empty pNBU2 vector did not, demonstrating that BT2086 is responsible for bile salt hydrolase activity in Bt. (**D**) Representative UPLC-MS traces showing that Bt WT deconjugates the murine primary bile acid TβMCA but not TCA, whereas BTΔ2086 (Bt KO) does not deconjugate either bile acid.
DOI: https://doi.org/10.7554/eLife.37182.004

The following figure supplements are available for figure 2:

**Figure supplement 1.** Biological duplicates of percent deconjugation at 48 hr, glyco-conjugated bile acids.
DOI: https://doi.org/10.7554/eLife.37182.005

**Figure supplement 2.** Biological duplicates of percent deconjugation at 48 hr, tauro-conjugated bile acids.
DOI: https://doi.org/10.7554/eLife.37182.006

**Figure supplement 3.** Deconjugation heat maps, 24 hr timepoint.
DOI: https://doi.org/10.7554/eLife.37182.007

low to undetectable, while the levels of deconjugated βMCA could build up due to enterohepatic recirculation. We decided to focus our efforts on generating paired isogenic strains in one of these species, *B. thetaiotaomicron* (Bt). Although this strain displayed relatively weak TβMCA-

deconjugating activity, Bt had been previously shown to be amenable to genetic manipulation, allowing knockout of putative BSH genes (*Cullen et al., 2015*; *Koropatkin et al., 2008*).

We performed a BLASTP search of the characterized BSH from *C. perfringens* (*Ridlon et al., 2006*) against the Bt genome and identified two genes, BT2086 and BT1259, as putative BSH. We constructed unmarked deletions of these genes using allelic exchange and then tested the resultant mutants for their ability to deconjugate bile acids in whole cell culture using UPLC-MS. The BtΔ2086 mutant (henceforth referred to as Bt KO) had lost the ability to cleave conjugated bile acid substrates. In contrast, the BtΔ1259 mutant displayed no loss-of-function phenotype (*Figure 2C*). Complementation of the Bt KO strain with BT2086 restored BSH activity (*Figure 2C*), confirming that BT2086 is necessary for bile acid deconjugation in Bt. Since bile salt hydrolases and penicillin V amidases (PVA) both belong to the cholylglycine hydrolase (CGH) family and share a high degree of sequence homology, it is possible that BT1259 is a PVA, although additional experiments would be needed to definitively establish this activity (*Jones et al., 2008*; *Panigrahi et al., 2014*). Finally, we verified that when incubated with both TβMCA and TCA, Bt wild-type (Bt WT) deconjugated TβMCA but not TCA, whereas the Bt KO strain did not deconjugate either bile acid (*Figure 2D*).

## Bacteroidetes BSH exhibit evolutionary diversity

A phylogenetic grouping of the 20 Bacteroidetes strains assayed revealed that while the species that deconjugate bile acids based on the amino acid conjugate (Group II – gray, *Figure 2A*) form a partial clade (*Figure 3A*), the strains that exhibit selectivity based on the steroid core (Group I – red) and those that display no selectivity (Group III – blue) are not separated into distinct clades. A BLAST-P search using BT2086 as a query gene identified candidate BSH genes in 19 of the 20 Bacteroidetes strains tested. *Bacteroides finegoldii* DSM 17565 did not display BSH activity and also lacked a putative BSH. A phylogenetic tree resulting from the multiple sequence alignment of these 19 candidate BSH genes revealed a lack of homology among enzymes within a given activity group (*Figure 3B*). Group II enzymes, which had formed a clade at the strain level, are now separated into two groups, and steroid core-selective strains (Group I) do not cluster significantly. Taken together, these findings suggest that preference for C12 = H over C12 = OH primary bile acid cores is an activity that may have evolved multiple times independently from related members of the BSH superfamily.

## Genetic removal of Bt BSH results in specific changes to murine bile acid pools in vivo

To test our hypothesis that deleting a single bacterial gene, the bile salt hydrolase BT2086, would result in a predictable and selective alteration of the in vivo bile acid pools, GF mice were monocolonized with Bt WT or Bt KO (monocolonization experiment, *Figure 4A*). To further assess effects of this single microbial gene on overall host metabolism and energy utilization, we also performed an experiment in CLAMS (Comprehensive Lab Animal Monitoring System) cages using three groups of animals: (1) mice monoassociated with Bt WT, (2) mice monoassociated with Bt KO or (3) GF control mice which remained sterile (CLAMS experiment, *Figure 4A*). For both studies, over a 4-week period, mice were fed a high-fat, high-sugar diet designed to mimic a Western-style human diet (60% kcal% fat). For the last week of the CLAMS experiment, mice were transferred from gnotobiotic isolators to pre-sterilized metabolic cages with continuous monitoring in the CLAMS system in order to carefully monitor metabolic status.

We first confirmed that BT2086 was expressed in vivo by performing qRT-PCR on cecal contents from Bt WT-colonized mice (*Figure 4—figure supplement 1*). As expected, no BT2086 transcripts were detected in the cecal contents of BT KO-colonized mice. We then performed bile acid analyses on tissues and blood from mice in both experiments. As we predicted, Bt KO-colonized mice displayed higher levels of TβMCA in cecal contents than Bt WT-colonized mice in the monocolonization experiment (*Figure 4B*). Bt KO-colonized mice also exhibited significantly lower levels of βMCA (p<0.0001), the product of TβMCA hydrolysis, than Bt WT-colonized mice. Importantly, the levels of TCA remained unchanged between the two groups, and no CA was detected in either group. These results are consistent with our in vitro data showing that the Bt BSH can deconjugate C12 = H but not C12 = OH primary bile acids. We observed the same significant difference in βMCA levels in feces (*Figure 4C*, red highlight boxes). In the CLAMS experiment, in agreement with previous reports (*Sayin et al., 2013*), GF mice had significantly higher overall bile acid levels than colonized

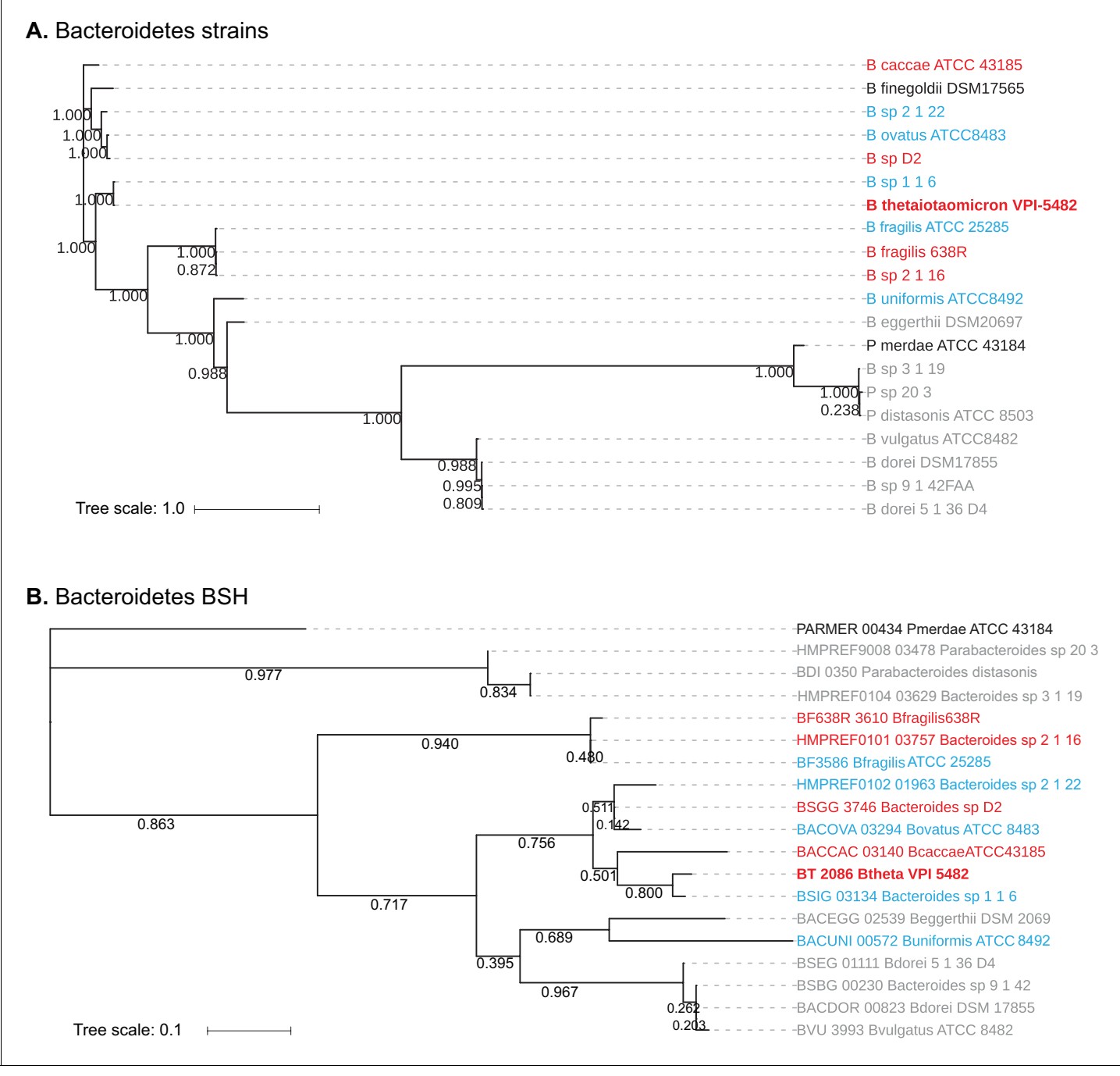

**Figure 3.** Homology-based classification of Bacteroidetes strains and putative BSH genes. (**A**) Phylogenetic tree of 20 Bacteroidetes strains using alignment-based whole proteome phylogeny (PhyloPhlAn). Bacteroidetes strains from Group II (gray; deconjugation based on amino acid) form a partial clade, while Group I (red) and Group III (blue) strains do not separate into distinct clades. (**B**) Phlyogenetic tree of candidate Bacteroidetes BSH genes. A search for BLAST-P matches of BT2086 identified an ortholog in 19 of the 20 Bacteroidetes species assayed. Numbers next to the branches represent the percentage of replicate trees in which this topology was reached in a bootstrap test of 1000 replicates. No significant clustering of Bacteroidetes strains into clades based on enzymatic activity was observed. Scale bars represent number of nucleotide substitutions per site.

DOI: https://doi.org/10.7554/eLife.37182.008

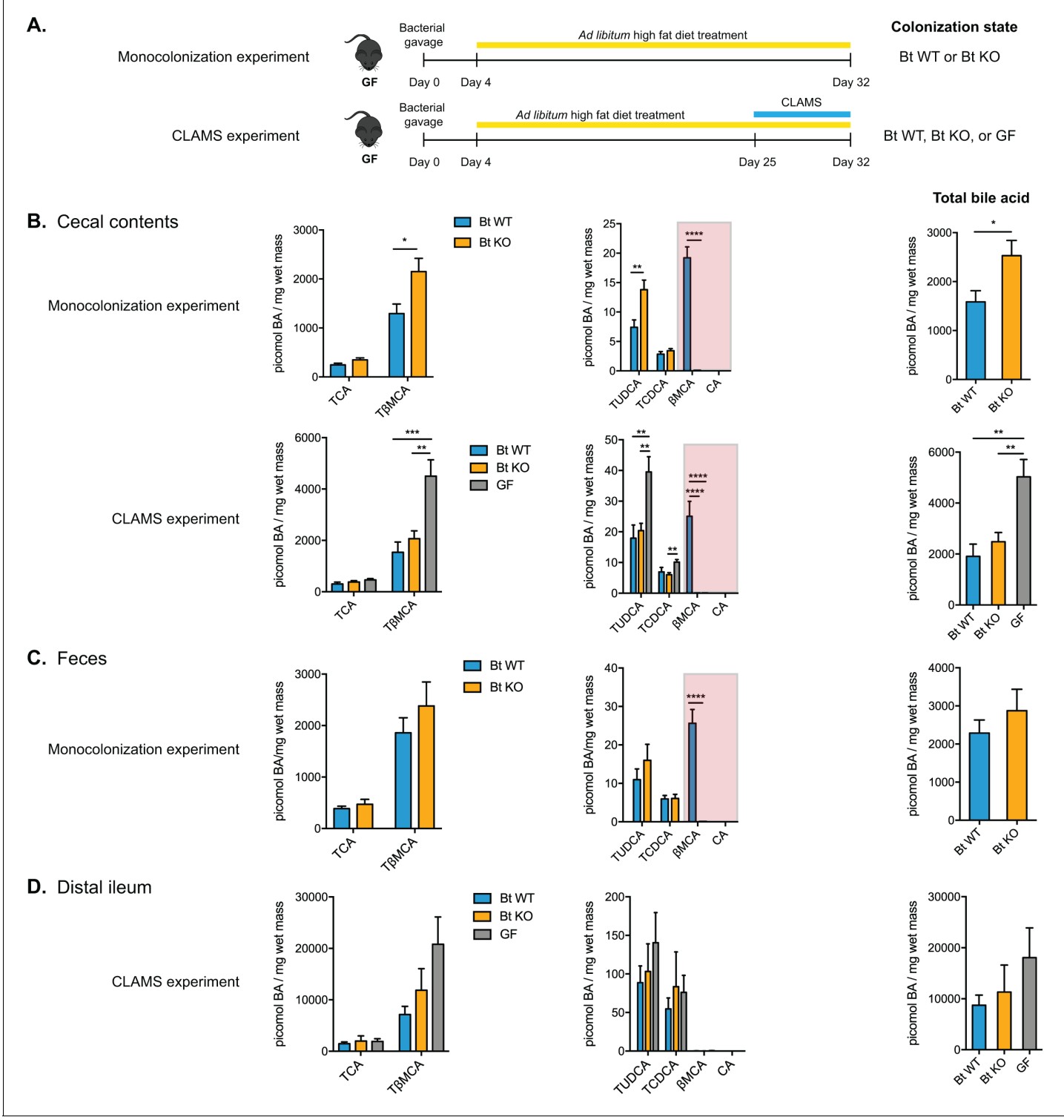

**Figure 4.** Monocolonization of GF mice with BT WT and Bt KO results in predictably altered bile acid pools. (**A**) Male germ-free C57BL/6 mice were monocolonized with either Bt WT or BT KO and fed a high-fat diet (HFD) for 4 weeks (monocolonization experiment). In a second experiment, monocolonized mice (Bt WT or Bt KO) or GF mice were fed a HFD for 4 weeks and transferred to CLAMS during the 4th week to monitor metabolic inputs and outputs (CLAMS experiment). (**B**) Bile acid profiling using UPLC-MS revealed that Bt KO-colonized mice displayed higher levels of TβMCA in cecal contents than Bt WT-colonized mice while the levels of TCA remained the same between the two groups. βMCA levels were significantly higher in Bt WT-colonized animals and no CA was detected in any group (red boxes). (**C–D**) Similar changes were observed in feces (**C**) and distal ileum (**D**), although no βMCA was observed in any group in the distal ileum. Data are presented as mean ± SEM. BA (bile acid). For the monocolonization

*Figure 4 continued on next page*

*Figure 4 continued*

experiment, n = 12 mice per group, Welch's t test. For the CLAMS experiment, n = 7–8 mice per group, one-way ANOVA followed by Tukey's multiple comparisons test. *p<0.05, **p<0.01, ***p<0.001, ****p<0.0001.

DOI: https://doi.org/10.7554/eLife.37182.009

The following figure supplements are available for figure 4:

**Figure supplement 1.** Confirmation of in vivo expression of Bt BSH.
DOI: https://doi.org/10.7554/eLife.37182.010

**Figure supplement 2.** Bile acid composition in the liver and plasma.
DOI: https://doi.org/10.7554/eLife.37182.011

mice (p=0.0012 Bt WT vs GF, p=0.0071 BT KO vs GF). Consistent with the monocolonization experiment, cecal contents of Bt KO-colonized mice displayed significantly lower levels of βMCA (p<0.0001, Bt WT vs GF and BT KO vs GF) than cecal contents of Bt WT-colonized mice, while CA remained undetectable in both groups (*Figure 4B*).

We also profiled the bile acid composition in the distal ileum, the site of active bile acid reuptake from the small intestine, in the CLAMS experiment. As expected, the bile acid concentrations were approximately fivefold higher in this compartment than in cecal and fecal contents (*Figure 4D*) (*Sayin et al., 2013*). We observed the same trend as we had noted in cecal contents, with higher TβMCA levels in Bt KO-colonized mice, although the differences were not statistically significant (p=0.9343). During sacrifice, we noted that the distribution of this food debris was not uniform along the length of the small intestine. This heterogeneity of contents in the distal ileum may help explain the large range of bile acid measurements observed in this compartment.

In contrast to the cecum, feces, and distal ileum, the liver and circulating plasma (*Figure 4—figure supplement 2*) of Bt WT- and Bt KO-colonized mice contained similar bile acid compositions, with no significant differences noted. These data are consistent with previous observations that the greatest differences between GF and conventionally raised mice were in the cecum and colon, not in the liver or the blood (*Sayin et al., 2013*). We also observed a significant upregulation of bile acid synthesis genes in the liver (vide infra), suggesting that de novo bile acid synthesis may lessen the observed differences between the two groups.

Taken together, our data show that we can rationally manipulate the in vivo bile acid pool in the cecum and to a lesser extent in the small intestine and distal colon (i.e. feces) using a *Bacteroides* BSH enzyme that selectively cleaves C12 = H but not C12 = OH conjugated primary bile acids. Importantly, this selective hydrolysis allows us to modulate the levels of TβMCA, a known FXR antagonist, while leaving the levels of TCA, an FXR agonist unchanged.

## Bt BSH status affects host metabolic indications

Having shown that Bt BSH status selectively determines composition of the bile acid pool in monocolonized GF mice, we next sought to explore how these specific changes in bile acid levels affected host metabolism. Strikingly, Bt KO-colonized mice gained less weight on the high-fat diet than Bt WT-colonized mice in the monocolonization experiment (*Figure 5A*). This result is notable because it has been shown that GF mice are more resistant to weight gain when fed a high-fat diet (*Bäckhed et al., 2007*). In addition, we performed a relatively short diet intervention compared to other studies that have used HFD to study metabolic changes (*Jiang et al., 2015*; *Joyce et al., 2014*; *Rao et al., 2016*; *Serino et al., 2012*), and we did not expect to observe significant changes in body weight over the course of a shorter experiment. Importantly, the host effects observed are not due to differences in colonization efficiency. In both experiments, Bt WT and Bt KO efficiently colonized the GI tract and remained the only bacterial species in the mono-associated animals (*Figure 5B* and *Figure 4—figure supplement 1*). These data suggest that the observed metabolic changes are rather due to alterations in the bile acid pool driven by the presence or absence of the Bt BSH.

Consistent with the reduced weight gain phenotype, we observed lower levels of triglycerides, cholesterol, and free fatty acids in plasma (*Figure 5C*) as well as lower triglyceride levels in liver (*Figure 5D*) of Bt KO-colonized compared to Bt WT-colonized mice in the monocolonization

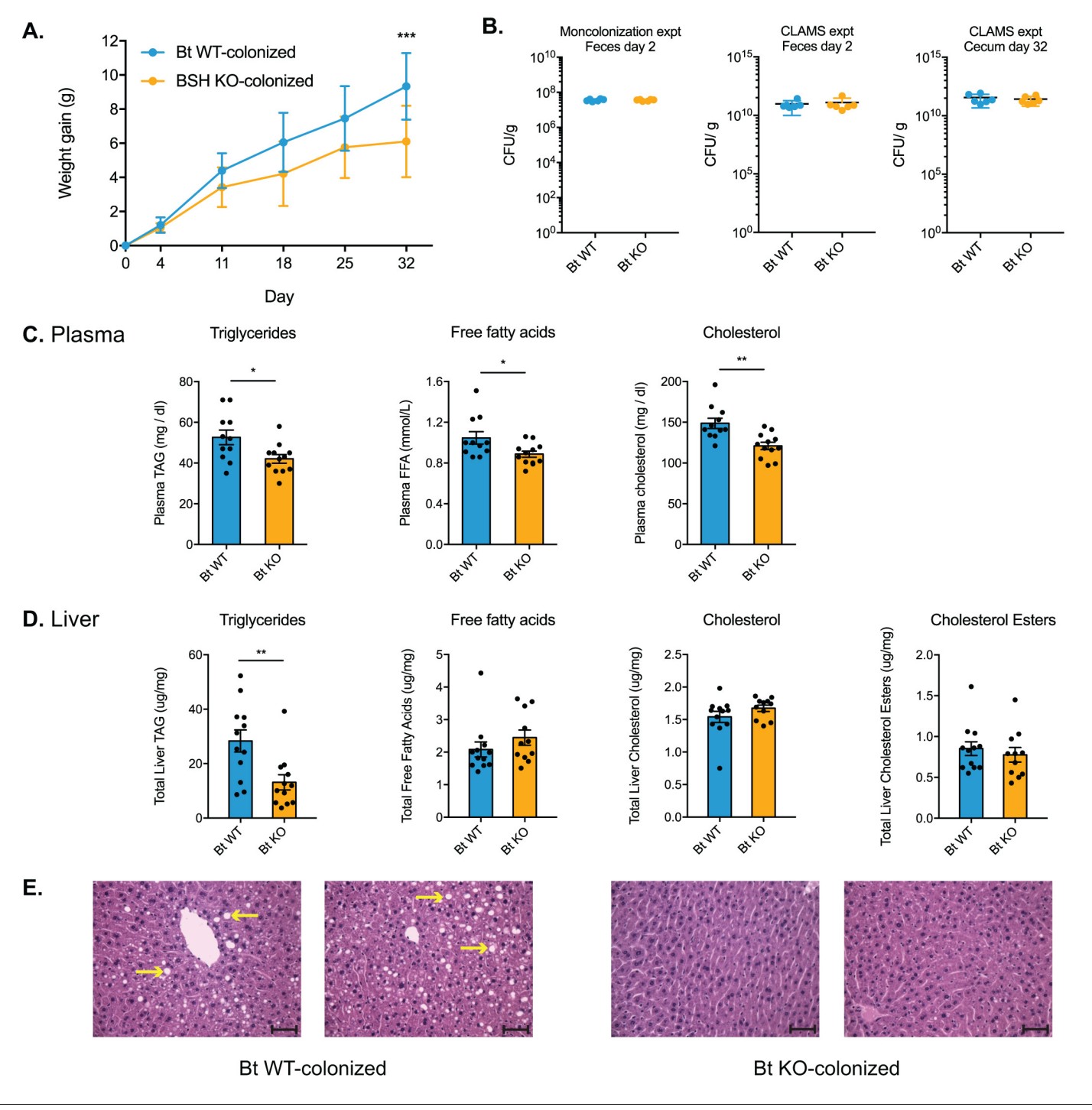

**Figure 5.** Bt BSH colonization status affects weight gain and lipid profiles. (A) Weight-matched, male, germ-free C57BL/6 mice monocolonized with Bt KO gained less weight during a 4-week diet challenge than Bt WT-colonized mice. n = 12 mice per group, multiple t test analysis using the FDR method (Q = 1%), ***p<0.001 (FDR-corrected). (B) Microbial biomass did not differ between the Bt WT and Bt KO strains in GF mice. For the monocolonization experiment, frozen feces (day 2 post-colonization) were plated to determine colony-forming units per gram. For the CLAMS experiment, fresh feces (day 2) were plated. Cecal contents collected at sacrifice (day 32) were also plated to confirm maintenance of monocolonized status throughout the experiment. No CFU were detected in the GF group. n = 6 samples per group, Mann-Whitney test. (C–D) Lipid levels in plasma and blood, monocolonization experiment. (C) Triglyceride, free fatty acid, and cholesterol levels were lower in the plasma of Bt KO-colonized mice. (D) Levels of triglycerides were reduced in Bt KO-colonized mice. No significant differences in liver free fatty acids, liver cholesterol, or liver cholesterol esters were observed. n = 12 mice per group, Welch's t test. (E) Representative H and E staining of liver sections (monocolonization experiment). Bt

*Figure 5 continued on next page*

*Figure 5 continued*

KO-colonized mice displayed decreased liver steatosis. Scale bars, 100 μm. Yellow arrows indicate representative white lipid droplets. n = 4 mice per group. All data are presented as mean ± SEM. *p<0.05, **p<0.01.

DOI: https://doi.org/10.7554/eLife.37182.012

experiment. Bt KO-colonized mice also exhibited less liver steatosis than Bt WT-colonized mice, consistent with the lower liver triglyceride levels in the former group (*Figure 5E*).

In order to further investigate the effects of Bt BSH status on host metabolism, we transferred Bt KO- or Bt WT-colonized or GF mice to metabolic cages (CLAMS experiment). After a 24 hr acclimation period, we monitored metabolic inputs and outputs for 6 days. We observed significant metabolic differences between the three groups of mice. Both Bt KO-colonized mice and GF mice displayed a lower respiratory exchange ratio (RER) than Bt WT-colonized mice (*Figure 6A*). RER is calculated as the ratio of carbon dioxide produced to oxygen consumed and is used as a measurement of the relative utilization of carbohydrates versus lipids as an energy source (carbohydrate utilization RER = 1, lipid RER = 0.7). Thus, our data indicate that both the Bt KO-colonized and GF mice

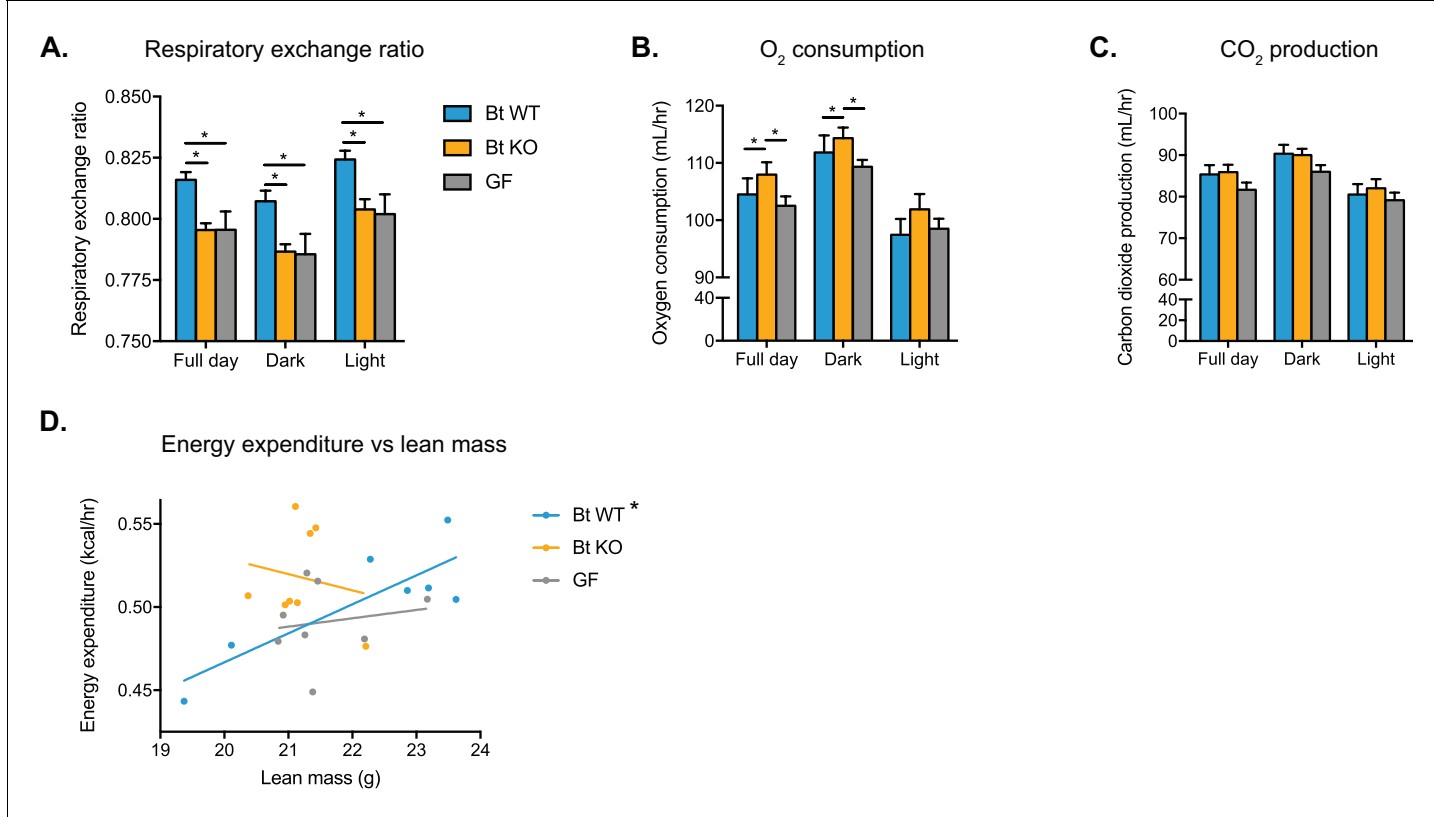

**Figure 6.** Bt WT-colonized, Bt KO-colonized, and GF mice display distinct metabolic phenotypes. Mice were monitored in metabolic cages during the final week of the CLAMS experiment and continued on a HFD. Mice were allowed to acclimate to cages for 24 hr prior to the start of data acquisition. (A) Respiratory exchange ratio (RER). One-way ANOVA followed by Tukey's multiple comparisons test, *p<0.05. (B–C) Oxygen consumption and (B) carbon dioxide production (C). ANCOVA with lean body mass as the covariate, *p<0.05. (D) Regression plot of energy expenditure as a function of lean mass. Energy expenditure (EE) is given by EE = CV x VO2, where CV = 3.815 + 1.232(RER). For the Bt WT-colonized group, *p=0.0168, indicating the slope is significantly non-zero. For the Bt KO-colonized and GF groups, p=0.6806 and p=0.6930, respectively. For (A–C), data are represented as mean ± SEM, where the number of samples is the number of mice. For Bt WT-colonized group, n = 7 mice, and for Bt KO-colonized and GF groups, n = 8 mice per group.

DOI: https://doi.org/10.7554/eLife.37182.013

The following figure supplement is available for figure 6:

**Figure supplement 1.** Locomotor activity.

DOI: https://doi.org/10.7554/eLife.37182.014

are utilizing more lipids for energy than carbohydrates relative to the Bt WT-colonized mice. While Bt KO-colonized mice consumed more oxygen than Bt WT-colonized mice (*Figure 6B*), there were no significant differences in carbon dioxide production between groups (full day, Bt WT vs Bt KO p=0.4041; Bt WT vs GF p=0.3239; Bt KO vs GF p=0.0606) (*Figure 6C*). These data are consistent with the lower RER observed in Bt KO-colonized mice. No statistically significant differences in loco-motor activity were noted between the three groups (*Figure 6—figure supplement 1*). We then used linear regression to investigate the relationship between metabolic rate and body weight in the three groups of mice. Conventionally raised mice as well as humans display a positive linear correlation between energy expenditure and body mass (*Fricker et al., 1989*; *Moruppa, 1990*). While Bt WT-colonized mice displayed this linear relationship (p=0.0168, $R^2$ = 0.7134), strikingly, both Bt KO-colonized (p=0.6806, $R^2$ = 0.03017) and GF (p=0.6930, $R^2$ = 0.02782) mice did not (*Figure 6D*). These data suggest that the deletion of a single bacterial gene, a selective bile salt hydrolase, results in loss of the relationship between metabolic rate and body weight in the host. Taken together, our data from both the monocolonized experiment and the CLAMS experiment suggest that the Bt KO-colonized mice exhibit a metabolic phenotype distinct from Bt WT-colonized mice.

## Distal ileum bile acid pools exhibit similar detergent properties

The reduced respiratory exchange ratio and weight gain of Bt KO-colonized mice suggest a reduced energy availability profile that is consistent with either reduced food consumption or less efficient caloric extraction from food. In the monocolonization experiment, Bt KO-colonized mice consumed less food during HFD feeding than Bt WT-colonized mice (−2.28 g ± 1.36 g vs. +3.39 g±1.61 g per cage per week, respectively, compared to weekly average for all cages, p=0.0165). This result indicates that decreased caloric intake may be a contributing factor in the former group's decreased weight gain. Since bile acids act as biological detergents that aid in digestion, it is conceivable that the differences in bile acid pools between the groups could alter caloric extraction efficiency. To test this hypothesis, we performed a detergent assay in which we determined the ability of the bile acid pools to solubilize a mixture of fats representative of lipolysis products in the small intestine (*Hofmann, 1963*). Bile acid pools for Bt KO- and Bt WT-colonized mice were reconstituted using the mean values for individual compounds measured in the distal ileum and incubated with a 1:1:1 mixture of oleic acid, sodium oleate and 1-oleoyl-*rac*-glycerol under conditions representative of those in the small intestine (150 mM NaCl, pH 6.3, 37°C) (*Hofmann, 1963*). Sodium dodecyl sulfate (SDS) was used as a positive control at its critical micelle concentration (8.2 mM). At both 5 hr and 24 hr time points, we did not detect any differences in solubilization at four different fat concentrations as measured by the turbidity of the resulting mixtures (*Figure 7A*). Taken together, these data suggest that the metabolic differences observed between the Bt WT- and Bt KO-colonized mice are not due to different detergent abilities of the bile acid pools. In further support of this conclusion, fecal bomb calorimetry did not reveal any differences in energy remaining in fecal pellets from Bt WT-colonized, Bt KO-colonized, or GF mice, indicating that there were no notable differences in caloric energy extraction from food between these groups (*Figure 7B*). These data suggest that the observed metabolic differences between Bt WT- and Bt KO-colonized mice may be due to differences in bile acids acting as signaling molecules in the host.

## Bt BSH status affects host global transcriptional response

In order to investigate the gene regulatory mechanisms underlying the metabolic changes observed in Bt KO- compared to Bt WT-colonized mice, we performed RNA-sequencing (RNA-Seq) on distal ileum from the monocolonization experiment (*Figure 8—figure supplement 1*). We decided to focus our analysis on the distal ileum for three reasons. First, while known bile acid receptors are highly expressed in both liver and intestinal tissue, we observed larger differences in bile acid pool composition in the GI tract (i.e. small intestine, cecum, feces) than the liver and blood, suggesting that differences in bile-acid-mediated signaling effects will likely be greater in the small intestine than in the liver. Second, bile acid concentrations are significantly higher in the small intestine than the cecum and colon, the other sites at which we observed differences in bile acid pool composition (approximately 5-fold and 100-fold higher, respectively) (*Sayin et al., 2013*). Third, following passage through the small intestine, bile acids are absorbed and recirculated back to the liver primarily

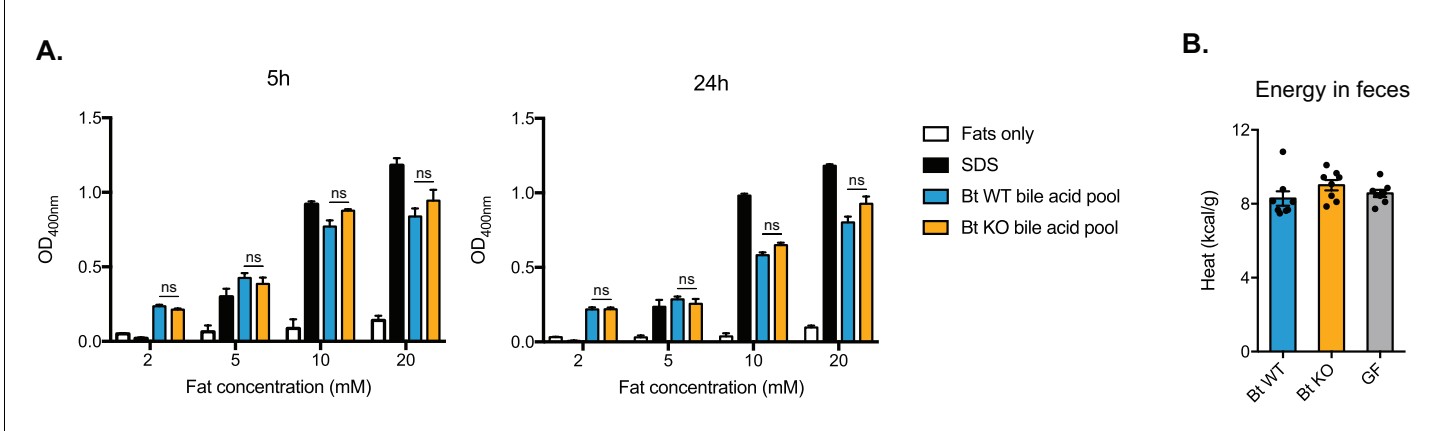

**Figure 7.** Detergent properties of ileal bile acid pools do not significantly differ. (**A**) Bile acid pools were reconstituted in vitro using the mean values measured from the distal ileum of Bt KO- and Bt WT-colonized mice (CLAMS experiment). Pools were added to four different concentrations of a mixture of fats representative of lipolysis products in the small intestine and incubated under physiologically relevant conditions (37°C, pH 6.3). SDS (sodium dodecyl sulfate) was used as a positive control at its critical micelle concentration (8.2 mM). No significant differences in solubilization effects as measured by OD400 were observed at 5 hr and 24 hr time points. n = 3 biological replicates per condition, one-way ANOVA followed by Tukey's multiple comparisons test. (**B**) No significant differences were observed in energy content of feces collected from Bt KO-colonized, Bt WT-colonized, or GF mice (CLAMS experiment). Feces were collected from CLAMS cages. n = 7–8 mice per group, one-way ANOVA followed by Tukey's multiple comparisons test. Each data point represents the mean of two calorimetry experiments per mouse. All data are presented as mean ± SEM.
DOI: https://doi.org/10.7554/eLife.37182.015

in the distal ileum (*Dawson et al., 2009*), making this site the nexus for bile acid sensing and transport in the GI tract.

Global transcriptional analysis of the distal ileum identified 12,432 genes, of which 428 genes were differentially expressed (adjusted FDR $\leq$ 0.05, fold-change $\geq \pm 1.5$) between the Bt KO- and Bt WT-colonized mice. Of those genes, the majority (314 genes) were increased in the Bt KO-colonized mice (*Figure 8A*). Multidimensional scaling analysis (MDS) revealed that the two monocolonized groups segregate based on their transcriptional profiles (*Figure 8B*). Gene Ontology (GO) and KEGG pathway analyses of RNA-Seq expression data revealed coordinated changes in gene expression related to metabolism, circadian rhythm, immune response, and histone modifications (*Figure 8C*).

The largest group of differentially expressed genes were those related to host metabolism. We observed significant changes in genes related to carbohydrate and lipid metabolism, amino acid degradation and nitrogen metabolism, and xenobiotic metabolism. In particular, genes involved in the transport (*Slc2a1*) and breakdown (*Hk1/2*, *Pfkl/m*) of glucose were upregulated, whereas *G6pc* (glucose-6-phosphatase), the final enzyme in the gluconeogenesis pathway, was significantly downregulated (8.8-fold), indicating a shift away from gluconeogenesis and toward glycolysis in the distal ileum of Bt KO-colonized mice. We confirmed the transcriptional change of *G6pc* in distal ileum using qPCR (*Figure 8D*). Consistent with these findings, we observed significantly higher blood glucose levels in Bt KO-colonized mice compared to Bt WT-colonized mice in the CLAMS experiment (p=0.0228), indicating an increase in glucose available for glycolysis in the distal ileum (*Figure 9A*).

The expression pattern for genes related to lipid metabolism was more complex, with pathways related to both lipogenesis and lipid breakdown upregulated in Bt KO-colonized animals. Two key genes in the ketogenesis pathway, *Bdh1* (3-hydroxybutyrate dehydrogenase 1) and *Hmgcs2* (3-hydroxy-3-methylglutaryl-CoA synthase 2), were significantly upregulated (*Figure 8C*), indicating an increase in the use of lipid and ketogenic amino acid degradation for energy production in the host. Additional genes related to amino acid degradation (*Hao2*, *Nos1*, *Pcca*, *Tat*) were also significantly upregulated. Expression of genes involved in the biosynthesis of both glycerophospholipids, in particular phosphatidic acid (*Dgkg*, *Dgkh*, *Gpam*, *Mboat1*, *Mboat2*), and sphingolipids, in particular cerebrosides and gangliosides (*Glb1*, *St3gal5*, *St6galnac6*, *Ugt8a*), was also higher in KO-colonized mice. Complex fats synthesized via de novo lipogenesis serve as ligands for PPAR type II nuclear receptors (*Lodhi et al., 2011*). RNA-Seq data revealed that *Pparg* expression was significantly up-

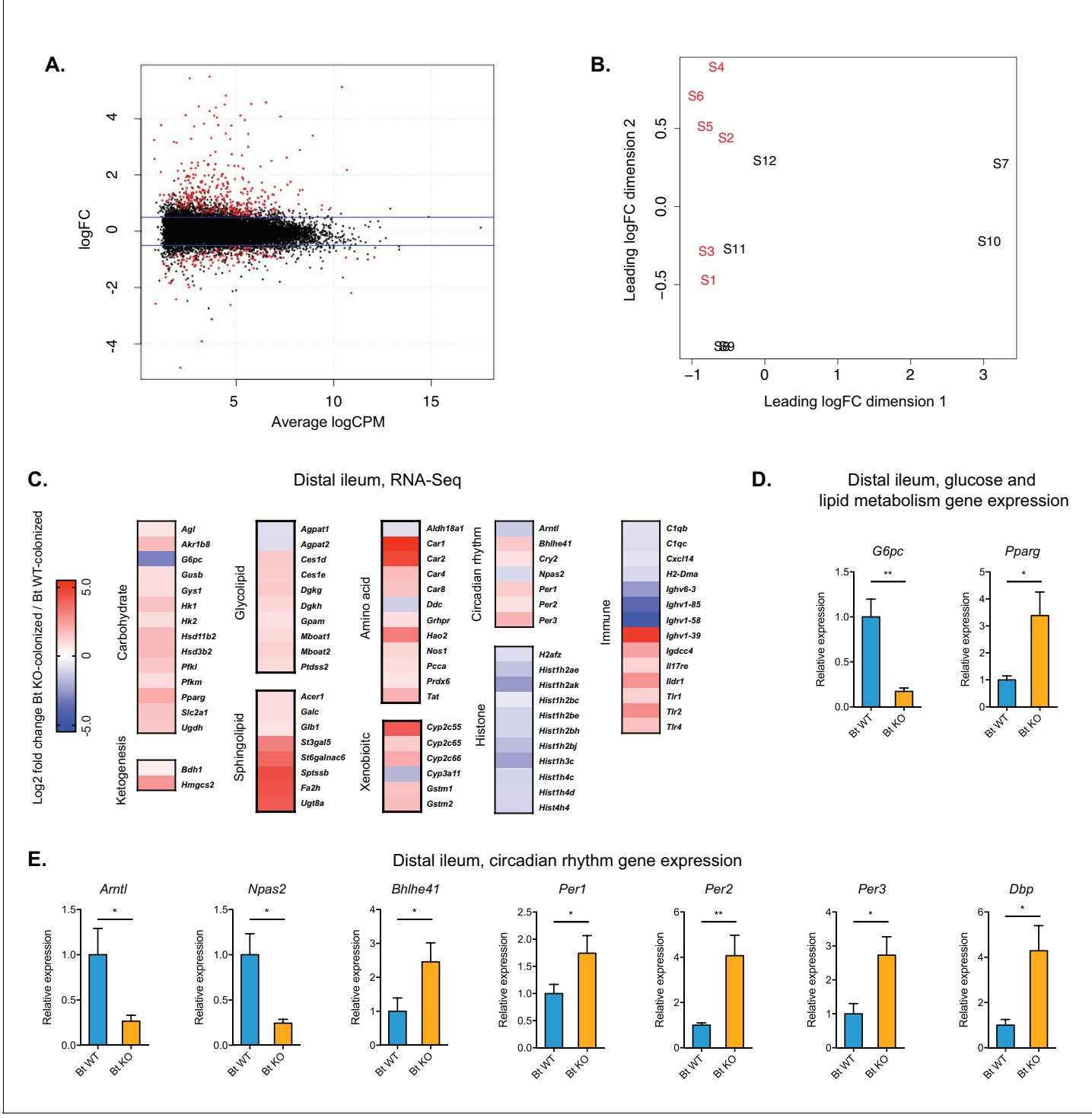

**Figure 8.** Global transcription analysis revealed changes in metabolism, circadian rhythm, immune response, and histone modification pathways in the distal ileum of Bt KO- vs. Bt WT-colonized mice (monocolonization experiment). (**A**) Log2-transformed fold change in normalized RNA-seq gene counts in the distal ileum of GF mice colonized with Bt KO relative to mice colonized with Bt WT. MA plot showing the relationship between average concentration (logCPM) and fold-change (logFC) across the genes. Each gene is represented by a black dot. Significant differentially expressed genes are colored in red. The blue lines represent logFC ±0.5 threshold. (**B**) Multidimensional scaling (MDS) plot of two monocolonized groups (S1-S6: Bt WT-colonized mice; S7-S12: Bt KO-colonized mice) derived from RNA-Seq normalized gene counts, showing that samples segregate based on colonization status (Bt WT vs. Bt KO). (**C**) Changes in the host distal ileum transcriptome between Bt WT- and Bt KO-colonized conditions. Heatmap shows statistically significant fold changes of genes identified using differential expression analysis (FDR ≤ 0.05 and absolute Log2 fold-change ≥0.5). n = 6
*Figure 8 continued on next page*

*Figure 8 continued*

samples for each group. (D–E) Gene expression in the distal ileum as measured by qPCR. Data are presented as mean ± SEM; n = 12 mice/group; *p<0.05, Welch's t test.

DOI: https://doi.org/10.7554/eLife.37182.016

The following figure supplements are available for figure 8:

**Figure supplement 1.** RNA-seq workflow.
DOI: https://doi.org/10.7554/eLife.37182.017
**Figure supplement 2.** RNA-seq, biological coefficient of variation.
DOI: https://doi.org/10.7554/eLife.37182.018

regulated in Bt KO-colonized mice (*Figure 8C*). Activation of *Pparg* has been shown to both enhance glucose metabolism and increase lipid uptake (*Martin et al., 1998*), consistent with our broader transcriptional analysis. We confirmed that *Pparg* expression was significantly upregulated in KO-colonized animals by qPCR (p=0.0207) (*Figure 8D*). Collectively, these data suggest that ileal cells in KO-colonized mice have shifted toward a regime of enhanced glycolysis and increased lipid uptake for the purposes of both the synthesis of complex fats and the breakdown of lipids for energy.

Transcriptional analysis also revealed changes in genes regulating circadian rhythm. The observed inverse relationship between expression of the transcriptional activators (*Npas2* and *Arntl,* decreased in Bt KO-colonized mice) and circadian repressors (*Per1*, *Per2*, *Per3*, *Cry2*, increased in Bt KO-colonized mice) is consistent with the transcription-translation negative feedback loop that establishes diurnal rhythms (*King and Takahashi, 2000*). The relative changes in circadian rhythm regulation genes were validated using qPCR (*Figure 8E*). These data indicate that tissues in the distal ileum of Bt KO-colonized mice exist in an altered circadian synchronization state compared to those of Bt WT-colonized mice. Genes involved in immune homeostasis and histone modifications were also differentially expressed. Of particular note, Toll-like receptors (*Tlr1*, *Tlr2*, *Tlr4*), innate immune receptors that play key roles in recognizing microbially produced molecules (*Akira et al., 2001*), were significantly upregulated in our Bt KO-colonized mice. Taken together, these data suggest that bile acid pool alteration elicited a broader scope of changes in the host beyond those directly related to energy production and lipid synthesis.

## Bile acid pools alter the expression of FXR-dependent and FXR-independent genes in the liver and distal ileum

We next sought to investigate the hypothesis that the two bile acid pools would differentially and predictably affect FXR signaling in the small intestine and the liver. Prior work has shown that the gut microbiome mainly affects FXR targets in the ileum but not the liver (*Sayin et al., 2013*). Specifically, activation of ileal FXR leads to production of fibroblast growth factor 15/19 (FGF15 in mice and FGF19 in humans). FGF15 then translocates to the liver where it binds to the FGFR4/β-Klotho complex and represses the expression of *Cyp7a1*, which encodes an enzyme catalyzing the rate-limiting step in bile acid synthesis from cholesterol (*Ding et al., 2015*). In this way, activation of FXR in the ileum downregulates bile acid synthesis in the liver. In our system, the levels of the FXR antagonist TβMCA were higher in the cecal contents of Bt KO- versus Bt WT-colonized mice, while the levels of the FXR agonist TCA remain constant between these two groups. Based on these results, we predicted that we would observe inhibition of FXR-dependent pathways in the distal ileum and perhaps the liver in Bt KO-colonized mice. We measured expression of FXR-dependent genes in these tissues using qPCR. Contrary to our expectations, we did not observe a significant difference in genes downstream of FXR, including *Nr0b2/Shp* (p=0.2018), *Fgf15* (p=0.6213), and *Fabp6/Ibabp* (p=0.6425), in the distal ileum (*Figure 9B*). We did observe a downregulation of *Nr0b2/Shp* and upregulation of *Cyp7a1* in the liver of Bt KO-colonized mice, results that are consistent with increased TβMCA-mediated FXR antagonism in Bt KO-colonized mice (*Figure 9C*). The total bile acid pool concentration in cecal contents was higher in Bt KO-colonized mice (*Figure 4A*), consistent with an increase in *Cyp7a1* transcription resulting in an increase in bile acid synthesis. We also observed decreases in the expression of other genes in the liver that are regulated by FXR, including *Apoc2*, which encodes a protein that is secreted into plasma and activates lipoprotein lipase, as well as increases in genes that are negatively regulated by the FXR target gene *Nr0b2/Shp*, including

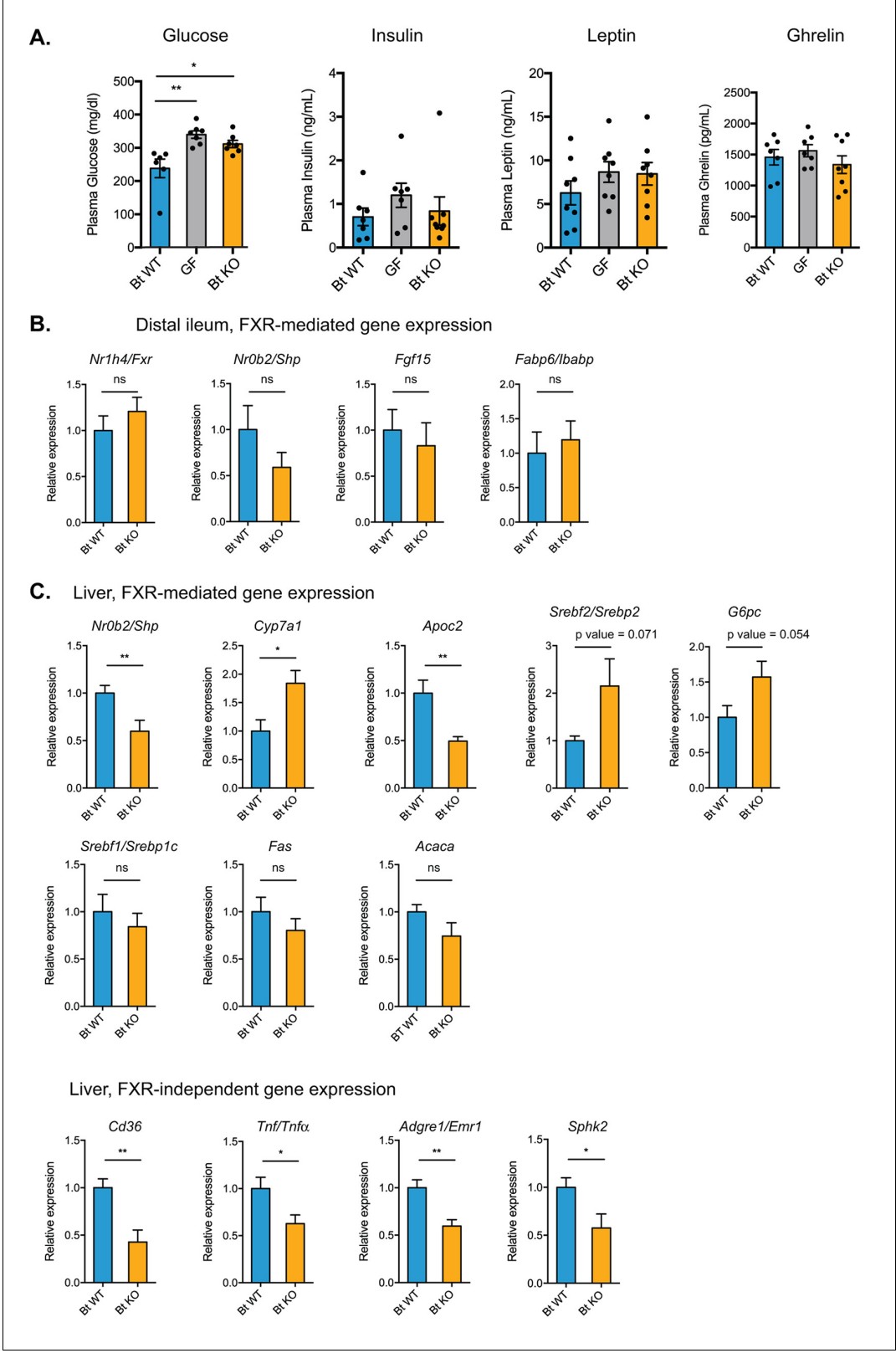

**Figure 9.** Bt BSH status affects transcription of genes in FXR-dependent and FXR-independent pathways. (**A**) Plasma glucose and hormone levels, CLAMS experiment. Mice were fasted for 4 hr prior to terminal blood draw. n = 7–8 mice per group, data are presented as mean ± SEM, one-way ANOVA followed by Tukey's multiple comparisons test, *p<0.05, **p<0.01. (**B**) Expression of genes in FXR-mediated pathways in the distal ileum was

*Figure 9 continued on next page*

*Figure 9 continued*

not significantly different between Bt KO- and Bt WT-colonized mice as measured by qPCR. (**C**) Gene expression in the liver as measured by qPCR. Genes in both FXR-mediated pathways (*Nr0b2/Shp, Cyp7a1, Apoc2*) and pathways not known to be mediated by FXR (*Cd36, Tnf/Tnfα, Adgre1/Emr1, Sphk2*) were significantly affected by Bt BSH status. (**B–C**) Data are presented as mean ± SEM; n = 12 mice/group; *p<0.05, **p<0.001, ns - not significant, Welch's t test.

DOI: https://doi.org/10.7554/eLife.37182.019

sterol regulatory element-binding protein 2 (*Srebf2/Srebp2*) and glucose-6-phosphatase (*G6pc*) (*Figure 9C*). While the former gene regulates cholesterol biosynthesis in the liver, the later gene catalyzes the final step in gluconeogenesis. The increase in *G6pc* in the liver of Bt KO-colonized mice is notable because this gene is significantly downregulated in the distal ilea of these mice (*Figure 8D*). Taken together, our data are consistent with a scenario in which bile-acid-mediated FXR antagonism is affecting pathways in the liver but not the ileum of Bt KO-colonized mice.

While some patterns of gene expression in the liver may be explained by FXR signaling, changes in the expression levels of certain notable pathways are not consistent with FXR-controlled regulation. We would expect to see an increase in the expression of the gene encoding sterol regulatory element binding protein 1 c (*Srebf1/Srebp1c*) as well as the downstream genes *Fas* and *Acc*, which are involved in de novo fatty acid synthesis, in the liver of Bt KO-colonized animals. No significant differences in expression of these genes, however, were observed between Bt KO- and WT-colonized mice (*Srebf1/Srebp1c*, p=0.5018; *Fas*, p=0.3292; *Acaca*, p=0.1302) (*Figure 9C*). In addition, we observed significant decreases in genes not known to be under the control of FXR, including *Cd36* (p=0.0015), a gene encoding a fatty acid transporter, the immune-related genes tumor necrosis factor alpha (Tnf/*Tnfα*, p=0.0225) and EGF-like module-containing mucin-like hormone receptor-like 1 (*Adgre1/Emr1*, p=0.0011), and the G-protein-coupled receptor S1pr2 target gene sphingosine kinase 2 (*Sphk2*, p=0.0274) (*Nagahashi et al., 2015*), in the liver of Bt KO-colonized mice (*Figure 9C*). These results indicate that other host receptors may be involved in the transcriptional changes and metabolic differences observed. Taken together, our data suggest that changing the in vivo bile acid pool using selective expression of a bacterial bile salt hydrolase results in significant alterations in host gene expression, and that these changes are due not to the detergent properties of bile acids but rather to their activities as signaling molecules.

## Discussion

In this work, we identified a group of gut strains from the bacterial phylum Bacteroidetes that exhibit selective bile salt hydrolase activity. These bacteria selectively hydrolyze conjugated bile acid substrates based on the hydroxylation pattern of the steroidal core as opposed to the amino acid conjugate. Since the majority of BSH characterized to date from Bacteroidetes and Firmicutes are promiscuous and do not display selective deconjugation activity based on the bile acid substrate, it is possible that selective BSH activity may be an evolved trait. The lack of distinct clustering of Group I (i.e. steroidal core-selective) BSH at both the strain and protein levels suggests this activity that may have arisen multiple times in evolutionary history from different bacterial hydrolase precursors. Structural comparisons of closely related BSH with different selectivity profiles may reveal individual amino acids that could be responsible for the activities observed. It is also possible that differential trafficking of either the bile acid substrate or product or of the BSH protein itself (*Begley et al., 2006*) in these Bacteroidetes strains may be responsible for some of the differences in reactivity. Additional microbiological, biochemical, and structural studies will be needed to answer these questions.

After identifying the gene responsible for BSH activity in *B. thetaiotaomicron* and generating a mutant (Bt KO), we leveraged these isogenic strains in order to manipulate the in vivo bile acid pool in a highly specific manner in monocolonized GF mice. Bt KO-colonized mice, which contained significantly higher cecal TβMCA levels than Bt wild type (WT)-colonized mice, gained less weight on a HFD, had lower liver and plasma lipid levels, and displayed a respiratory exchange ratio that was shifted toward lipid utilization. These changes in host metabolism are particularly striking in light of the fact that the only difference between these two groups of mice was the presence or absence of

a single bacterial gene. Remarkably, the presence of this BSH gene in BT WT-colonized mice was able to recover the positive linear correlation between energy expenditure and lean body mass normally observed in both conventional mice and humans (*Fricker et al., 1989*; *Moruppa, 1990*). This result suggests that specific genes in the gut microbiome may contribute to the establishment of host phenotypes not previously considered to be affected by the resident microbiota.

At a transcriptional level, genes related to metabolic pathways, circadian rhythm, immune modulation, and histone modifications were significantly altered in Bt KO- compared to Bt WT-colonized mice. Since TβMCA is a known FXR antagonist, we expected to observe changes in host gene expression that were consistent with downregulation of FXR-mediated pathways in Bt KO-colonized mice. The decreased expression of FXR target genes *Nr0b2/Shp* and *Apoc2* as well as the increased expression of *Cyp7a1*, the rate-limiting enzyme in bile acid biosynthesis, are consistent with a regime of FXR antagonism in the livers of Bt KO- compared to Bt WT-colonized mice. These transcriptional changes suggest that the observed increase in total bile acids in the cecal contents of Bt KO-colonized mice is due to FXR-dependent bile acid biosynthesis in the liver.

Other phenotypic and transcriptional differences observed between Bt KO- and Bt WT-colonized mice are not readily explained by FXR antagonism, however. Conventionally colonized FXR knockout (*Nr1h4$^{-/-}$*) mice display less weight gain on a high-fat diet than wild-type mice (*Prawitt et al., 2011*) and also have decreased liver expression of *Nr0b2/Shp* and increased expression of *Cyp7a1* (*Sayin et al., 2013*), consistent with our results in Bt KO-colonized mice. *Nr1h4$^{-/-}$* mice, however, exhibit increased triglyceride and cholesterol levels in plasma (*Cariou et al., 2006*; *Lambert et al., 2003*; *Sinal et al., 2000*) and low blood glucose and delayed intestinal glucose absorption when fasted (*Cariou et al., 2006*; *van Dijk et al., 2009*). Bt KO-colonized mice displayed the opposite phenotypes, including increased glucose and decreased triglyceride and cholesterol levels in plasma and a shift toward increased expression of glucose uptake and utilization genes in the distal ileum when fasted. While the expression of the FXR target genes *Nr0b2/Shp*, *Fgf15*, and *Fabp6/Ibabp* in the ileum are decreased in *Nr1h4$^{-/-}$* mice (*Sayin et al., 2013*), we observed no transcriptional differences in these genes in Bt KO- and Bt WT-colonized mouse ilea. In addition, while the expression of hepatic gluconeogenesis genes is decreased in FXR-deficient mice (*Cariou et al., 2005*; *Duran-Sandoval et al., 2005*; *Ma et al., 2006*), we observed an increase in the expression of glucose-6-phosphatase (*G6pc*) in the liver of Bt KO-colonized mice. Taken together, these comparisons may indicate that many of the phenotypic and transcriptional differences noted in BT KO-colonized mice are either FXR-independent or not directly dependent on FXR-mediated signaling.

These results raise the possibility, then, that bile acid signaling through other host receptors may be in part responsible for the observed differences in host metabolism. Returning to the RNA-Seq data, we noted that there were significant differences in the expression of ileal genes involved in xenobiotic metabolism in Bt KO-colonized compared to BT WT-colonized mice. The pregnane X receptor (PXR) has been shown to play a central role in the response to xenobiotics, and in particular, in the transcriptional regulation of cytochrome P450 3A (*Cyp3A*) genes (*Bertilsson et al., 1998*). The expression of *Cyp3a11*, a mouse gene known to be regulated by PXR (*Kliewer et al., 1998*), was significantly decreased (3.7-fold) in Bt KO-colonized animals. This result indicates that PXR-dependent pathways may be suppressed in these mice compared to Bt WT-colonized mice.

PXR also plays an important role in glucose and lipid homeostasis and energy metabolism (*Gao and Xie, 2010*; *Kodama et al., 2004*; *Kodama et al., 2007*; *Nakamura et al., 2007*). PXR knockout (*Nr1i2$^{-/-}$*) mice gain less weight on a high-fat diet than wild-type mice and also display decreased liver steatosis and hepatic triglyceride levels (*Spruiell et al., 2014*). Importantly, in contrast to FXR knockout (*Nr1h4$^{-/-}$*) mice, *Nr1i2$^{-/-}$* mice fed a high-fat diet exhibit increased fasting blood glucose levels and unchanged fasting insulin levels compared to wild-type mice (*He et al., 2013*; *Spruiell et al., 2014*). These metabolic phenotypes are consistent with those observed in Bt KO-colonized mice. Finally, PXR has been shown to be necessary and sufficient for the activation of the fatty acid transport gene *Cd36* in the liver (*Zhou et al., 2006*), and we observed a decrease in hepatic *Cd36* expression in BT KO-colonized mice. Taken together, our data are consistent with a regime of reduced PXR activation in Bt KO-colonized mice and perhaps suggest that PXR signaling may be involved in some of the metabolic phenotypes observed.

We cannot rule out the possibility that host receptors beyond FXR and PXR may be involved in the differences noted between Bt KO- and Bt WT-colonized mice. Exploration of bile acids as modulators of host metabolic, circadian rhythm, and immune response via binding to nuclear receptors

and GPCRs is an experimental trajectory that warrants further investigation. Moreover, although our results support the conclusion that the observed changes in host metabolism are mediated by signaling properties of bile acids and not by their detergent activities, it is not yet clear which gene-level changes are driving the organism-level metabolic effects. Although our biochemical, transcriptional, and CLAMS data are consistent with a regime of decreased food intake in Bt KO- compared to Bt WT-colonized mice, we did not observe significant differences in plasma levels of leptin (p=0.4648, Bt WT vs Bt KO) and ghrelin (p=0.7783, Bt WT vs Bt KO), hormones regulating satiety and hunger (*Figure 8C*). Additional studies are needed to explore the contributions of bile acid signaling to host energy expenditure and feeding behavior.

Finally, because mice and humans possess different primary bile acids, there is the question of whether the observed changes in both bile acid composition and host metabolism are relevant for humans. While the major primary bile acids in mice are TβMCA and TCA (*Sayin et al., 2013*), humans produce glyco- and tauro-conjugated CDCA and CA (*Russell, 2003*). Our in vitro results show that selective *Bacteroides* strains cleave conjugated C12 = H (e.g. βMCA, CDCA) but not C12 = OH (e.g. CA) primary bile acids. Based on our in vivo results, one would predict that these strains would cleave conjugated CDCA while leaving conjugated CA untouched in the human gut. Furthermore, analysis of data from the first and second phases of the Human Microbiome Project has revealed that the composition of the human gut community, specifically species of Bacteroidetes, is highly personalized. While Firmicutes were more temporally variable within individuals, Bacteroidetes species, and in particular the *Bacteroides* genus, displayed mainly inter-individual variation (*Kraal et al., 2014*; *Lloyd-Price et al., 2017*). Our results suggest that *Bacteroides* species status in individuals may in part determine downstream bile acid pool composition in these people. Finally, the FXR pathway as well as other host receptor pathways that may act as bile acid targets are highly conserved in mammals (*Reschly et al., 2008*), suggesting that discoveries about fundamental host signaling in mice are also likely to be operable in humans. Future studies in mice with humanized bile acid pools may reveal how selective *Bacteroides* BSH activity is likely to affect metabolism in the human host.

# Materials and methods

**Key resources table**

| Reagent type (species) or resource | Designation | Source or reference | Identifiers | Additional information |
|---|---|---|---|---|
| Strain, strain background (*Bacteroides thetaiotaomicron*) | VPI 5482 [CIP 104206T, E50, NCTC 10582] | ATCC | ATCC 29148 | |
| Strain, strain background (*Bacteroides caccae*) | VPI 3452A [CIP 104201T, JCM 9498] | ATCC | ATCC 43185 | |
| Strain, strain background (*Bacteroides ovatus*) | [NCTC 11153] | ATCC | ATCC 8483 | |
| Strain, strain background (*Bacteroides vulgatus*) | [NCTC 11154] | ATCC | ATCC 8482 | |
| Strain, strain background (*Bacteroides uniformis*) | Not applicable | ATCC | ATCC 8492 | |
| Strain, strain background (*Parabacteroides distasonis*) | [NCTC 11152] | ATCC | ATCC 8503 | |
| Strain, strain background (*Bacteroides fragilis*) | VPI 2553 [EN-2; NCTC 9343] | ATCC | ATCC 25285 | |
| Strain, strain background (*Parabacteroides merdae*) | VPI T4-1 [CIP 104202T, JCM 9497] | ATCC | ATCC 43184 | |
| Strain, strain background (*Bacteroides eggerthii*) | Not applicable | DSMZ | DSM-20697 | |
| Strain, strain background (*Bacteroides finegoldii*) | 199 | DSMZ | DSM-17565 | |
| Strain, strain background (*Bacteroides dorei*) | 175 | DSMZ | DSM-17855 | |

*Continued on next page*

*Continued*

| Reagent type (species) or resource | Designation | Source or reference | Identifiers | Additional information |
|---|---|---|---|---|
| Strain, strain background (*Bacteroides dorei*) | 5_1_36/D4 | BEI | HM-29 | |
| Strain, strain background (*Bacteroides* sp.) | 1_1_6 | BEI | HM-23 | |
| Strain, strain background (*Bacteroides* sp.) | 9_1_42FAA | BEI | HM-27 | |
| Strain, strain background (*Bacteroides* sp.) | D2 | BEI | HM-28 | |
| Strain, strain background (*Bacteroides* sp.) | 3_1_19 | BEI | HM-19 | |
| Strain, strain background (*Bacteroides* sp.) | 2_1_16 | BEI | HM-58 | |
| Strain, strain background (*Parabacteroides* sp.) | 20_3 (Deposited as *Bacteroides* sp., Strain 20_3) | BEI | HM-166 | |
| Strain, strain background (*Bacteroides* sp.) | 2_1_22 | BEI | HM-18 | |
| Strain, strain background (*Bacteroides fragilis*) | 638R | Other | | Gift from Seth Rakoff-Nahoum, Boston Children's Hospital |
| Strain, strain background (*Lactobacillus plantarum*) | NCIMB 8826 [Hayward 3A, WCFS1] | ATCC | BAA-793 | |
| Strain, strain background (*Clostridium perfringens*) | NCTC 8237 [ATCC 19408, CIP 103 409, CN 1491, NCIB 6125, NCTC 6125, S 107] | ATCC | ATCC 13124 | |
| Strain, strain background (*Escherichia coli* S17-1 λ pir) | *E. coli* S17-1 λ pir | Other | | Gift from Michael Fischbach, Stanford University |
| Strain, strain background (*B. thetaiotaomicron* VPI-5482 Δtdk) | Bt WT | Other | | Gift from Michael Fischbach, Stanford University |
| Strain, strain background (*B. thetaiotaomicron* VPI-5482 ΔtdkΔ2086) | BtΔ2086 | This paper | | See Materials and methods, 'Construction of *Bacteroides thetaiotaomicron* knockout mutants' |
| Strain, strain background (*B. thetaiotaomicron* VPI-5482 ΔtdkΔ1259) | BtΔ1259 | This paper | | See Materials and methods, 'Construction of *Bacteroides thetaiotaomicron* knockout mutants' |
| Strain, strain background (*B. thetaiotaomicron* VPI-5482 Δtdk Δ2086 pNBU2_erm_us1311_BT2086) | BtΔ2086,2086+ | This paper | | See Materials and methods, 'Construction of *Bacteroides thetaiotaomicron* complementation strains' |
| Strain, strain background (*B. thetaiotaomicron* VPI-5482 Δtdk Δ2086 pNBU2_erm_us1311_CTRL) | BtΔ2086,CTRL+ | This paper | | See Materials and methods, ' Construction of *Bacteroides thetaiotaomicron* complementation strains' |
| Recombinant DNA reagent | pExchange-tdk (plasmid) | PMID: 18611383 | | |
| Recombinant DNA reagent | pNBU2_erm_us1311 (plasmid) | PMID: 25574022 | | |
| Sequence-based reagent (knockout primer pairs) | BT2086_UF | Eurofins Genomics | | GAA AGA AGA TAA CAT TCG AGT CGA CAT CCA AAC CCA GTG TGA ACT |
| Sequence-based reagent (knockout primer pairs) | BT2086_UR | Eurofins Genomics | | CAT ATT ACT TCC AAA TTA AAT AGT TGA TAC TC |

*Continued on next page*

*Continued*

| Reagent type (species) or resource | Designation | Source or reference | Identifiers | Additional information |
|---|---|---|---|---|
| Sequence-based reagent (knockout primer pairs) | BT2086_DF | Eurofins Genomics | | GAG TAT CAA CTA TTT AAT TTG GAA GTA ATA TGT AGT CGA TAG TTA GTT ATG TGG TAA G |
| Sequence-based reagent (knockout primer pairs) | BT2086_DR | Eurofins Genomics | | CCA CCG CGG TGG CGG CCG CTC TAG AAG CAG ACG TTA TCC TGG TTT C |
| Sequence-based reagent (knockout primer pairs) | BT1259_UF | Eurofins Genomics | | GAA AGA AGA TAA CAT TCG AGT CGA CGG ATG ATT ATT GCC CCA TTT TG |
| Sequence-based reagent (knockout primer pairs) | BT1259_UR | Eurofins Genomics | | CGT ACA CAT AAT TTC GAT TTT TAG TTA TAG |
| Sequence-based reagent (knockout primer pairs) | BT1259_DF | Eurofins Genomics | | CTA TAA CTA AAA ATC GAA ATT ATG TGT ACG TAA ATT GAT AGC AGC TTG CTG C |
| Sequence-based reagent (knockout primer pairs) | BT1259_DR | Eurofins Genomics | | CCA CCG CGG TGG CGG CCG CTC TAG ACG TTT TTC TAC CGG ACG AAT C |
| Sequence-based reagent (complementation primer pairs) | us1311-BT-For-NdeI | Eurofins Genomics | | GGG TCC ATA TGA AGA AAA AAC TTA CGG GTG TTG C |
| Sequence-based reagent (complementation primer pairs) | BT-Rev-XbaI | Eurofins Genomics | | CTA GTC TAG ACT ACA TCA CCG GAG TTT CGA A |
| Sequence-based reagent (diagnostic primer) | pExchange_seq_UF | Eurofins Genomics | | CGG TGA TCT GGC ATC TTT CT |
| Sequence-based reagent (diagnostic primer) | pExchange_seq_DR | Eurofins Genomics | | AAC GCA CTG AGA AGC CCT TA |
| Sequence-based reagent (diagnostic primer) | BT2086_seq_F1 | Eurofins Genomics | | CAA CTG TCC GGG TGA ATA TAA AG |
| Sequence-based reagent (diagnostic primer) | BT2086_seq_F2 | Eurofins Genomics | | GAA GTT TTC GTT GGG TGA ATG |
| Sequence-based reagent (diagnostic primer) | BT1259_seq_F1 | Eurofins Genomics | | AGA AGG TAC ATC GCC TGT AC |
| Sequence-based reagent (diagnostic primer) | BT1259_seq_F2 | Eurofins Genomics | | TAC TAT TCA CGC ACC ACA CC |
| Sequence-based reagent (diagnostic primer) | pNBU2_UNIV-F | Eurofins Genomics | | TAA CGG TTG TGG ACA ACA AG |
| Sequence-based reagent (diagnostic primer) | pNBU2_UNIV-R | Eurofins Genomics | | CAC AAT ATG AGC AAC AAG GAA TCC |
| Sequence-based reagent (qRT-PCR primer) | qBTBSH_F | Eurofins Genomics | | GCGTGCGGGACACAATAAAG |
| Sequence-based reagent (qRT-PCR primer) | qBTBSH_R | Eurofins Genomics | | TAGCCTGTTGCGATTACGCT |
| Sequence-based reagent (qRT-PCR primer) | qBT16s_F | Eurofins Genomics | | GTGAGGTAACGGCTCACCAA |
| Sequence-based reagent (qRT-PCR primer) | qBT16s_R | Eurofins Genomics | | CTGCCTCCCGTAGGAGTTTG |

## Reagents

Conjugated and unconjugated bile acids were purchased from Steraloids Inc. (Newport, RI). Oleic acid, sodium oleate and 1-oleoyl-*rac*-glycerol were purchased from Sigma Aldrich.

## Bacterial culturing

All Bacteroidetes strains were cultured at 37°C in brain heart infusion agar (Bacto) supplemented with with 5 mg/L hemin, 2.5 uL/L Vitamin K, and 500 mg/L cysteine HCl (BHI+). All strains were cultured under anaerobic conditions using an anaerobic chamber (Coy Laboratory Products) with a gas mix of 5% hydrogen and 20% carbon dioxide (balance nitrogen). *Escherichia coli* strains were grown aerobically at 37°C in LB medium supplemented with ampicillin to select for the pExchange-tdk plasmid.

## Bioinformatic search for candidate BSH in *B. thetaiotaomicron*

A BLASTP search was performed on Integrated Microbial Genomes, the US Department of Energy's Joint Genome Institute (IMG JGI) using the bile salt hydrolase from *C. perfringens* (NCBI Protein accession code WP_003461725) as the query sequence, with a cutoff expectation value of $1 \times 10^{-5}$.

## Construction of *B. thetaiotaomicron* knockout mutants

Plasmids and primers are listed in the Key Resources Table. All mutants were created in the *B. thetaiotaomicron* VPI-5482 Δ*tdk* background. The *BtΔ2086* and *BtΔ1259* mutants were constructed using a previously described counterselectable allelic exchange method (*Koropatkin et al., 2008*). Briefly, ~1 kb fragments upstream and downstream of the BT2086 and BT1259 genes were cloned and fused using primer pairs (BT2086KO UF/UR and DF/DR; BT1259KO UF/UR and DF/DR) and ligated into the suicide vector pExchange-*tdk*. The resulting vectors were electroporated into *Escherichia coli* S17-1 λ *pir* and then conjugated into *B. thetaiotaomicron*. Single-crossover integrants were selected on BHI-blood agar plates containing 200 μg/ml gentamicin and 25 μg/mL erythromycin, cultured in TYG medium overnight, and then plated onto BHI-blood agar plates containing 200 μg/ml 5-fluoro-2-deoxyuridine (FUdR). Candidate BT2086 and BT1259 deletions were screened by PCR using the diagnostic primers listed in the Key Resources Table and confirmed by DNA sequencing to identify isolates that had lost the gene.

## Construction of *B. thetaiotaomicron* complementation strains

The *B. thetaiotaomicron* complementation strains were constructed using a previously described method with slight modifications. Assembled construct designs were based on the mobilizable *Bacteroides* element NBU2 (*Wang et al., 2000*). Briefly, BT2086 was PCR-amplified, cloned as an NdeI/XbaI fragment into the constitutive expression vector *pNBU2_erm_us1311*, which contains the 300 bp region upstream of BT1311 (σ70), and transformed into *E. coli* S17-1 λ *pir* chemically competent cells (*Cullen et al., 2015*; *Degnan et al., 2014*). *E. coli* S17 lambda pir containing the desired plasmid or the *pNBU2_erm_us1311* control vector were cultured aerobically in 5 mL of LB media at 37°C, and the *Bacteroides* recipient strain (BtΔ2086) was cultured anaerobically in 5 mL BHI+ media at 37°C. The *E. coli* S17 donor strains and *B. theta* recipient strain were then subcultured in 5 mL of fresh media. At mid to late log growth, the transformed S17-1 cells were spun down, resuspended with *Bacteroides* strain (BtΔ2086) culture in 1 mL BHI+ media, spreaded on to a BHI+ 10% horse blood agar plate, and incubated aerobically at 37°C agar side down. After 16–24 hr, bacterial biomass from the conjugation plates was scraped and resuspended in 5 mL of BHI+ media and spread on to a BHI-blood agar plate containing 200 μg/mL gentamycin and 25 μg/mL erythromycin. Colonies were confirmed via PCR and sequencing using the diagnostic primers listed in the Key Resources Table. Recovery of function of the complementation strain was confirmed via UPLC-MS with 100 μM TUDCA as substrate.

## Phylogenetic analysis of candidate BSHs and Bacteroidetes strains

BLASTP searches were performed on Integrated Microbial Genomes (IMG JGI) using BT2086 as the query sequence, with a cutoff expectation value of $1 \times 10^{-5}$. Putative Bacteroidetes BSHs were identified from 19 of the 20 Bacteroidetes strains tested. A multiple sequence alignment was calculated using MUSCLE (*Edgar, 2004*). A phylogenetic tree was then computed from this alignment using PhyML (*Guindon et al., 2010*), choosing the LG substitution model, the SPR and NII (best) tree improvement method, 10 random starting trees, and bootstrap with 1000 replicas. The phylogenetic tree was visualized using iTOL (*Letunic and Bork, 2011*). A phylogenetic analysis of 20 Bacteroidetes strains was performed using PhyloPhlAn (*Segata et al., 2013*). All 20 fully or partial sequenced

microbial genomes were retrieved from IMG and the National Center for Biotechnology Information (NCBI).

## In vitro assays for bile acid deconjugation by Bacteroidetes

All strains were cultured in 4 mL of BHI+ medium overnight. The following day they were diluted to pre-log phase ($OD_{600}$ = 0.1) in fresh BHI+ to a final volume of 4 mL. Stock solution of taurine conjugated bile acids (TCA, TDCA, TCDCA, TUDCA, TLCA and TβMCA) or glycine conjugated bile acids (GCA, GDCA, GCDCA, GUDCA and GLCA) were added to each culture to obtain a final concentration of 50 μM of each bile acid. Cultures were then incubated in the anaerobic chamber at 37°C for 48 hr. At the 24 hr and 48 hr time points, 2 mL of each culture was extracted via the method described under 'Bile Acid Analysis - Sample Preparation for Bacterial Culture'.

## Gnotobiotic mouse experiments

Germ-free C57BL/6 mice were maintained in gnotobiotic isolators at the Massachusetts Host-Microbiome Center under a strict 12 hr light cycle and a constant temperature (21 ± 1°C) and humidity (55–65%). All experiments were conducted on 8–9 week old male mice. GF status prior to the gavage was confirmed on a bi-weekly basis microbiologically through culturing mouse stool on broad-spectrum plates in both aerobic and anaerobic conditions, as well as Gram staining homogenized mouse stool in 1xPBS. TSA Blood Agar plates were used for aerobic conditions, while Brucella Blood Agar plates were used for anaerobic conditions. Monocolonization or GF status following gavage was confirmed by plating of fecal pellets (described below) and by 16S rRNA gene PCR and 16S sequencing. All experiments involving mice were performed using IACUC approved protocols under Brigham and Women's Hospital Center for Comparative Medicine.

For the monocolonization experiment, 12 weight-matched mice per group were colonized with either *B. thetaiotaomicron* wild type (Bt WT) or the Δ2086 mutant (Bt KO) by oral gavage of overnight cultures as previously described (*Marcobal et al., 2011*). Frozen feces (day 2 post-colonization) were plated to determine CFU/g. The mice were co-housed in their respective groups in gnotobiotic isolators for the entire duration of the experiment. The mice were fed a standard diet containing 24% of calories from fat, 23% from protein, and 53% from carbohydrates (Autoclavable Mouse Breeder Diet 5021; LabDiet) for the first 4 days after gavage. After a 4-day acclimation period post-gavage, the mice were switched to a high-fat diet (Research Diets D12492) with 60 kcal% of fat sterilized by 10–20 kGy of gamma-irradiation. Mouse feces were collected 2 days after colonization to check bacterial colonization efficiency. This was achieved by homogenizing 1–2 fecal pellets in 1 mL PBS and then plating out 1:10 serial dilutions of the homogenate on BHI+ agar plates in the anaerobic chamber. BSH enzyme activities in different experimental groups were also checked via UPLC-MS for validation purposes. Fecal samples were collected on days 2, 4, 11, 18, 25 and 32 post-colonization and frozen at −80°C prior to analysis. Mice were fasted for 4 hr prior to sacrifice, at which point tissues and blood were collected.

For the CLAMS experiment, GF mice were colonized as above with a third group of age- and weight-matched GF mice used as an additional control group (eight mice per group). Fresh feces (day 2 post-colonization) were plated to determine CFU/g. On day 24 and day 25 post-colonization, mice were transported in pre-sterilized CLAMS cages to Brigham and Women's Hospital (BWH) Metabolic Core facility to conduct metabolic studies. Animals were housed individually in metabolic chambers maintained at 22°C under a 12 hr light/dark cycle with a constant access to food and water and maintained on a high-fat diet (Research Diets D12492). One mouse from the Bt WT-colonized group was excluded from the study because this animal refused food, lost 35% of its initial body weight in the CLAMS, and displayed GI tract abnormalities during sacrifice. Whole body metabolic rate was measured using the Oxymax open-circuit indirect calorimeter, Comprehensive Lab Animal Monitoring System (CLAMS, Columbus Instruments). Body composition was examined with Echo MRI (Echo Medical Systems, Houston, Texas) using the 3-in-1 Echo MRI Composition Analyzer (*Kazak et al., 2017*; *Mina et al., 2017*), and respiratory exchange ratio (RER), calorific value (CV), and energy expenditure (EE) are calculated by the equations below:

$$RER = \frac{VCO_2}{VO_2}$$

$$CV = 3.815 + 1.232*RER$$

$$EE = CV*VO_2$$

During sacrifice, whole blood was collected into commercially available EDTA-coated tubes (Milian Dutscher group). Cells were removed from plasma by centrifugation for 15 min at 2,000 g at 4°C. Plasma was transferred to a new eppendorf tube from the supernatant and stored in −80°C for further investigation. Cecal contents were collected at sacrifice (day 32) using sterile tools on a sterile field and plated to confirm maintenance of monocolonized or GF status throughout the experiment. No CFU were detected in the GF group. RNA extracted from these cecal contents was used for qRT-PCR using 16 s rRNA and *Bt BSH* primers (Key Resources Table, *Figure 4—figure supplement 1*).

Plasma insulin, glucose and ghrelin levels were analyzed by the Vanderbilt Mouse Metabolic Phenotyping Center. Plasma glucose was measured by a glucose oxidase method using an Analox Instruments GM9 glucose analyzer (Stourbridge, UK). Plasma insulin was measured by radioimmunoassay (Millipore). Total ghrelin was measured by radioimmunoassay (Millipore). Plasma leptin, glucagon levels were analyzed by ELISA kits (Crystal Chem). Total plasma cholesterol, triglyceride and free fatty acids (FFA) were measured by standard enzymatic assays, and liver tissues were extracted (*Folch et al., 1957*) and analyzed by the Vanderbilt University Metabolic Phenotyping Center (VUMC) using GC.

## Bile acid analysis

### Reagents
Stock solutions of all bile acids were prepared by dissolving compounds in molecular biology grade DMSO (Sigma Aldrich). These solutions were also used to establish standard curves. GCA and βMCA or GCDCA were used as the internal standard for GF mouse experiments and in vitro bacterial culture (for glyco-conjugated or tauro-conjugated bile acid), respectively. Solvents used for preparing UPLC samples were HPLC grade.

### Sample Preparation for Bacterial Culture
For extraction of bile acids, 2 mL of a bacterial culture was acidified to pH = 1 using 6N HCl. The culture was then extracted twice using 2 mL of ethyl acetate. In case of an emulsion, the biphasic solution was centrifuged at 2,500 g for 3 min to obtain a clear separation. The combined organic extracts were then dried over a $Na_2SO_4$ cotton plug, air dried, and reconstituted in 50% MeOH in $dH_2O$ for UPLC-MS analysis.

### Sample Preparation for Serum and Tissues
Bile acids from serum and tissues that were collected from GF mouse experiments were extracted using the method of Sayin *et. al* with the following modifications (*Sayin et al., 2013*). Serum was transferred to a 1.5 mL eppendorf tube. 100 μL of bile acid standard with 1 μM GCA (internal standard) dissolved in MeOH was added to the tube. After vortexing for 1 min, the sample was cooled to −20°C for 20 min. The sample was then centrifuged for 10 min at 15,000 g, and the supernatant (~100 μL) was transferred to another 1.5 mL eppendorf tube containing 50 μL of 50% MeOH in $dH_2O$. The sample was then centrifuged for another 10 min at 15,000 *g*, and 50 μL of the supernatant was transferred to a mass spec vial and injected onto the UPLC-MS. Dilutions were applied if the concentrations of certain bile acids were out of the maximum detection range of the standard curve.

Tissue samples (approximately 100 mg) were pre-weighed in homogenizing tubes (Precellys lysing kit tough micro-organism lysing VK05 tubes) with ceramic beads. 400 uL MeOH containing 10 uM internal std (GCA) was added and thereafter homogenized (5000 speed for 90 s*2, 6500 speed for 60 s, sample kept on ice between two runs) and spun down for 20 min at 15,000 *g*. Of the supernatant, 200 μL was then transferred to a tube containing 200 μL of 50% Methanol in $dH_2O$ followed by centrifugation for an additional 5 min at 15,000 *g*. Of the supernatant, 50 μL was used transferred to a mass spec vial and injected onto the UPLC-MS. Dilutions were applied if the concentrations of

certain bile acids were out of the maximum detection range of the standard curve. For quantifying bile acids, a mixture of bile acid standard pool was always carried out along with the experiment.

## UPLC-MS analysis

UPLC-MS was performed using a published method (*Swann et al., 2011*) with modifications outlined as follows. 1 µL of a 200 µM solution of extracted bile acids was injected onto a Phenomenex 1.7 µm, C18 100 Å, 100 × 21 mm LC colum at room temperature and was eluted using a 30 min gradient of 75% A to 100% B (A = water + 0.05% formic acid; B = acetone + 0.05% formic acid) at a flow rate of 0.350 mL/min. Samples were analyzed using an Agilent Technologies 1290 Infinity II UPLC system coupled online to an Agilent Technologies 6120 Quadrupole LC/MS spectrometer in negative electrospray mode with a scan range of 350–550 m/z (other mass spec settings: fragmentor - 250, gain - 3.00, threshold - 150, Step size - 0.10, speed (u/sec) - 743). Capillary voltage was 4500 V, drying gas temperature was 300℃, and drying gas flow was 3 L/min. Analytes were identified according to their mass and retention time. For quantification of the analytes, standard curves were obtained using known bile acids, and then each analyte was quantified based on the standard curve and normalized based on the internal standards. The limits of detection for individual bile acids are as follows: βMCA, 0.03 picomol/µL in serum or 0.1 picomol/mg wet mass in tissues; TβMCA, 0.01 picomol/µL, 0.04 picomol/mg wet mass; CA, 0.04 picomol/µL, 0.17 picomol/mg wet mass; TCA, 0.01 picomol/µL, 0.04 picomol/mg wet mass; UDCA, 0.04 picomol/µL, 0.16 picmol/mg wet mass; TUDCA, 0.01 picmol/µL, 0.04 picmol/mg wet mass; CDCA, 0.04 picmol/µL, 0.14 picomol/mg wet mass; TCDCA, 0.01 picomol/µL, 0.03 picomol/mg wet mass.

## Liver histological analysis

Histology of the liver samples for steatosis was performed using a reported method (*Rao et al., 2016*) in the Harvard Rodent Histopathology Core. Briefly, a portion of liver sample was cut and formalin-fixed, trimmed, cassetted and embedded in paraffin and stained with hematoxylin and eosin. Liver histology was assessed for steatosis on blinded sections.

## Fecal bomb calorimetry

To determine the remaining caloric content in the mouse feces, bomb calorimetry was carried out using the Parr Oxygen Bomb equipped with a Parr 6725 Semimicro Calorimeter module and aParr 6772 Calorimetric Thermometer module at the Brigham and Women's Hospital (BWH) Metabolic Core facility. Briefly, 30–100 mg of pooled fecal samples from the sample mice were dehydrated at 60℃ for 48 hr in a micro centrifuge tube. Calculated heats (cal/g) take into account diurnal variations in fecal output as well as any contaminants that had entered into the sample.

## Detergent assay

### Synthesis of the sodium salt of β-muricholic acid

To a solution of β-muricholic acid (6.0 mg, 0.0147 mmol) in 1.47 mL of methanol:toluene (1:1), 1M sodium hydroxide solution (0.15 µL, 0.016) was added and the resulting mixture was heated to 60℃ for 18 hr. The reaction mixture was then cooled to room temperature and concentrated using a rotovap. The mixture was azeotroped thrice using toluene to ensure removal of water and the residue was dried thoroughly before being used in the assay.

The detergent ability of bile acid pools was investigated using a modified fat solubility assay described by Hofmann and coworkers (*Hofmann, 1963*; *Lillienau et al., 1992*). Briefly, stock solutions of respective fats and bile acids of known concentrations were made by dissolving them in methanol. The fats were then mixed in 1:1:1 ratio in a 96-well plate to obtain the required final concentrations. Respective amounts of different bile acid stock solutions were then added to each of the wells to reconstitute the bile acid pool concentrations observed in vivo (Bt WT bile acid pool: 8.7 mM total, including 7.1 mM TβMCA; Bt KO bile acid pool: 14.0 mM total, including 11.9 mM TβMCA). For comparison, the detergent sodium dodecyl sulfate (SDS) was used as a positive control at its critical micellar concentration (8.2 mM). SDS was added as a solution in methanol to obtain the required final concentration. The plate was then dried overnight. The dried residue was then suspended in freshly prepared 0.15 M sodium phosphate at pH 6.3. In order to account for the slight difference in the concentration of $Na^+$ ions arising from the difference in concentrations of the bile

acids in the two pools, exogenous sodium chloride was added to maintain similar concentrations. The plate was then sealed and incubated at 37°C for 24 hr after which the absorbance was measured at 400 nm using a SpectraMax Plus 384 Microplate Reader spectrophotometer. At the lowest fat concentration (2 mM), this assay was performed in 1 mL of 0.15 M sodium phosphate in a 1 mL cuvette in order to obtain accurate $OD_{400}$ measurements. The solutions of fats and detergents were prepared in a similar manner as described above.

## Gene expression analysis

### RNA-Seq analysis

Total RNA was extracted from frozen mouse distal ileum obtained from the monocolonization experiment using RNeasy Mini Kit (Qiagen) and DNA was removed by on-column DNase digestion using RNAse-free DNAse Set (Qiagen) (n = 6). RNA quality was checked on Bioanalyzer before proceeding to library preparation and RNA-sequencing (paired-end, 100 bp read length) on Illumina HiSeq 2500 platform by the Biopolymers Facility at Harvard Medical School. Illumina's Ribo Zero H/M/R kit was used to perform ribosomal reduction. Agilent Tapestation 4200 was used for post-prep QC with the High Sensitivity D1000 assay. The data from the Tapestation assay was combined with the data from KAPA library quantification qPCR on the Applied Biosystems QuantStudio seven instrument. The libraries were pooled equimolar and were sequenced on the HiSeq 2500 at 8.0pM with 5% PhiX.

Reads were assessed for quality using FastQC and aggregated in MultiQC (Cambridge, UK: Babraham Institute, 2011, n.d.; *Ewels et al., 2016*). STAR aligner was used against mouse geneome GRCm38 revision 91 (*Dobin et al., 2013*). The edgeR package was performed for the differential expression analysis, using the exacTest calculation and Benjamini-Hochberg correction (FDR) (*McCarthy et al., 2012*; *Robinson et al., 2010*). FDR $\leq$ 0.05, fold-change (FC) $\geq \pm 1.5$ were set as the threshold. The goseq package was used to perform Gene Ontology (GO) and Kyoto Encyclopedia of Genes and Genomes (KEGG) pathway analyses, with significant differentially expressed genes subjected to a probability weighting function and gene length bias accounted (*Young et al., 2010*). RNA-Seq data are deposited in the Gene Expression Omnibus (GEO) database (accession GSE112571).

### qRT-PCR

Total RNA was extracted from liver and distal ileum tissues obtained from mice from the monocolonization experiment as previously described and reverse transcribed using a High-Capacity cDNA Reverse Transcription kit (Applied Biosystems). The resultant cDNA was diluted and analyzed by qRT–PCR using LightCycler 480 SYBR Green I Master (Roche). Reactions were performed in a 384-well format using a LightCycler 480 System (Roche) at Harvard Medical School's ICCB-Longwood Screening Facility. The $2^{-\Delta\Delta Ct}$ method (*Livak and Schmittgen, 2001*) was used to calculate relative changes in gene expression and all results were normalized to the mouse ribosomal protein L32 mRNA.

## Statistical analysis

Unless otherwise indicated in the figure legends, differences between experimental groups or conditions were evaluated using unpaired two-tailed Welch's t test for pairwise comparison, one-way ANOVA for multiple comparisons. Significance was determined as p value < 0.05. Statistical analysis and plotting for metabolic studies was performed in the R programming language with CalR, a custom package for analysis of indirect calorimetry using analysis of covariance with a graphical user interface (*Mina et al., 2017*).

## Acknowledgements

Mouse experiments were performed with the generous support of the Harvard Digestive Diseases Center Core D, supported by grant P30DK034854. We thank Vladimir Yeliseyev and Dimitrije Cabarkapa for technical support and advice. We acknowledge the VUMC Lipid Core for plasma and liver lipid profiles (grant # DK59637) and Vanderbilt University Medical Center Hormone Assay and Analytical Services Core for plasma ghrelin, insulin and glucose assays (NIH grants DK059637 and DK020593). We thank the Dana-Farber/Harvard Cancer Center in Boston, MA, for the use of the

Rodent Histopathology Core, which provided Paraffin and Frozen Sectioning and Tissue Trimming, Cassetting, Processing and Embedding service, supported in part by a NCI Cancer Center Support Grant # NIH 5 P30 CA06516. We thank Dr. Roderick T Bronson for histological analysis. We thank the BPF Next-Gen Sequencing Core Facility at Harvard Medical School for their expertise and instrument availability in support of Hiseq sequencing. We gratefully acknowledge Kristina Holton for help with RNA-Seq data analysis. We also thank HMS Research Computing and HMS O2 High Performance Compute Cluster, which is maintained by HMS Research Computing. We are indebted to Chih-Hao Lee and members of the Clardy group for helpful discussions. We thank Seth Rakoff-Nahoum and Michael Fischbach for use of material resources. This work was supported by an Innovation Award from the Center for Microbiome Informatics and Therapeutics at MIT, a grant from Harvard Digestive Diseases Center (supported by NIH grant 5P30DK034854-32), a Karin Grunebaum Cancer Research Foundation Faculty Research Fellowship (ASD), and a Wellington Postdoctoral Fellowship (LY).

## Additional information

### Competing interests

Sarah Craven Seaton: is currently affiliated with Indigo Agriculture, but the research was conducted when she was a Research Associate at Harvard Medical School. The author has no other competing interests to declare. A Sloan Devlin: is a consultant for Kintai Therapeutics. The other authors declare that no competing interests exist.

### Funding

| Funder | Grant reference number | Author |
|---|---|---|
| National Institutes of Health | P30DK034854 | Lina Yao<br>Sarah Craven Seaton<br>Sula Ndousse-Fetter<br>Arijit A Adhikari<br>Nicholas DiBenedetto<br>Lynn Bry<br>A Sloan Devlin |
| Wellington Postdoctoral Fellowship | | Lina Yao |
| Massachusetts Institute of Technology | Center for Microbiome Informatics and Therapeutics – Innovation Award | Arijit A Adhikari<br>A Sloan Devlin |
| Karin Grunebaum Cancer Research Foundation | Junior Faculty Award | A Sloan Devlin |

The funders had no role in study design, data collection and interpretation, or the decision to submit the work for publication.

### Author contributions

Lina Yao, Conceptualization, Data curation, Formal analysis, Investigation, Methodology, Writing—original draft, Writing—review and editing; Sarah Craven Seaton, Conceptualization, Data curation, Formal analysis, Investigation, Methodology, Writing—review and editing; Sula Ndousse-Fetter, Data curation, Formal analysis, Investigation; Arijit A Adhikari, Data curation, Formal analysis, Investigation, Methodology, Writing—original draft; Nicholas DiBenedetto, Data curation, Methodology, Writing—review and editing, Gnotobiotic Core staff member, Set-up and analysis for monocolonization and CLAMS mouse experiments; Amir I Mina, Data curation, Formal analysis, Methodology, Writing—review and editing, Metabolism Core staff member, Set-up and analysis for CLAMS mouse experiment; Alexander S Banks, Formal analysis, Supervision, Methodology, Writing—review and editing, Metabolism Core director, Planning and analysis for CLAMS mouse experiment; Lynn Bry, Supervision, Methodology, Writing—review and editing, Gnotobiotic Core director, Planning and analysis for monocolonization and CLAMS mouse experiments; A Sloan Devlin, Conceptualization,

Data curation, Formal analysis, Supervision, Funding acquisition, Investigation, Methodology, Writing—original draft, Project administration, Writing—review and editing

## Author ORCIDs

A Sloan Devlin 🆔 http://orcid.org/0000-0001-5598-3751

## Ethics

Animal experimentation: This study was performed in strict accordance with the recommendations in the Guide for the Care and Use of Laboratory Animals of the National Institutes of Health. All experiments involving mice were performed using IACUC approved protocols (Protocol # 2017N000053) at the Brigham and Women's Hospital Center for Comparative Medicine.

## Decision letter and Author response

Decision letter https://doi.org/10.7554/eLife.37182.025
Author response https://doi.org/10.7554/eLife.37182.026

## Additional files

### Supplementary files

• Supplementary file 1. Mouse experiment analyses.
DOI: https://doi.org/10.7554/eLife.37182.020

• Transparent reporting form
DOI: https://doi.org/10.7554/eLife.37182.021

### Data availability

RNA-Seq data are deposited in the Gene Expression Omnibus (GEO) database (accession GSE112571). All other data generated or analyzed during this study are included in the manuscript and supporting files.

The following dataset was generated:

| Author(s) | Year | Dataset title | Dataset URL | Database, license, and accessibility information |
|---|---|---|---|---|
| Yao L, Devlin AS | 2018 | A selective gut bacterial bile salt hydrolase alters host metabolism | www.ncbi.nlm.nih.gov/geo/query/acc.cgi?acc=GSE112571 | Publicly available at the NCBI Gene Expression Omnibus (accession no: GSE112571). |

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
