## [Decision Letter]

Thank you for submitting your article "A selective gut bacterial bile salt hydrolase alters host metabolism" for consideration by *eLife*. Your article has been reviewed by Wendy Garrett as the Senior Editor, a Reviewing Editor, and three reviewers. The following individuals involved in review of your submission have agreed to reveal their identity: Jason Ridlon (Reviewer #2). The reviewers have discussed the reviews with one another and the Reviewing Editor has drafted this decision to help you prepare a revised submission.

Summary:

All three reviewers were very enthusiastic about this manuscript and are in principle supportive of its publication pending revision in the upcoming *eLife* microbiome special issue. In this report, Yao et al. screened *Bacteroides* spp. for bile salt hydrolysis activity against a panel of conjugated bile salts and decided on *B. thetaiotaomicron* which deconjugated TβMCA (FXR antagonist) but not TCA (FXR agonist). Since *B. theta* has a well-defined genetic system, the authors knocked out the gene encoding bile salt hydrolase (BSH). Finally, they performed studies of gnotobiotic mice mono-associated with wildtype *B. theta* or an isogenic mutant lacking a candidate BSH gene. The two colonization groups exhibited altered metabolism in many aspects, suggesting that the ability of gut microbes to carry out this activity could impact host biology. This model is a great starting point for a more detailed dissection of the role of specific bacterial genes and metabolites in diet-induced obesity.

Essential revisions:

While there are many potential areas for future study, the reviewers agreed that one major issue that is necessary to address prior to publication is that the authors do not complement any of their mutations. It is unclear whether the genetic deletions cause phenotypes due to the absence of the targeted gene, polar effects on other genes, or random mutations that occurred elsewhere in the genome during the genetic engineering process. While it could be argued that such events are unlikely, complementation is an important part of any bacterial genetic analysis.

Additional caveats can likely be addressed in the Discussion section without the need for more experiments. This includes the unclear relevance of these findings to humans. The focus is on a primary bile acid (TβMCA) that is unique to mice and mono-colonized mice are obviously an artificial system. The authors should include more discussion about why human gut bacteria would have evolved enzymes that target mouse bile acids; perhaps consistent with some degree of enzyme promiscuity. The in vivo mechanism of action remains unclear, necessitating future studies. The authors should comment on the degree to which these phenotypes may be dependent or independent of FXR and discuss alternative mechanisms of action. The authors should also include a more in-depth discussion of the identified *Bacteroides* BSH and the potential mechanism that determines its specificity. Are there any distinguishing characteristics of selective versus non-selective strains, either in terms of primary sequence, genetic neighborhood, or other factors?

The screen presented in Figure 2 is a major part of the paper but needs to be presented in more detail. It was not clear why 25% was used as the threshold instead of presenting the actual% conversion. No time curves are shown making it hard to evaluate if 48 hours represents steady state for these reactions. There are also no replicates shown making it impossible to determine the reproducibility of these results. It also seems important to test the full panel of compounds against the isogenic strains to enable a more definitive test of selectivity. This experiment should be done in replicate and include appropriate statistics (traces aren't sufficient). The authors should also address the major caveat that *B. theta* was actually a poor metabolizer of Tβ MCA in Figure 2A, which is the focus of all the subsequent experiments.

Finally, the manuscript needs a careful round of revisions. Multiple inconsistent results and non-significant "trends" can be trimmed down substantially. The host phenotype sections were particularly difficult to read and interpret due to the mixing of non-significant and significant results. It was also hard to keep track of which results were reproduced across both mouse experiments, which could be addressed by a supplemental table or summary diagram.

Reviewer #1:

Yao et al. present an analysis of bile salt hydrolase activity by human gut *Bacteroides*. The major advance here is that unlike in other bacteria *Bacteroides* encode hydrolases selective for specific types of conjugated bile acids. They then go on to identify the gene responsible based on sequence homology to well-characterized enzymes, to confirm this gene by genetic deletion in *B. theta*, and to show that this gene is relevant in vivo using mono-association studies in gnotobiotic mice. This model is a great starting point for a more detailed dissection of the role of specific bacterial genes and metabolites in diet-induced obesity.

A major limitation to the study, as mentioned by the authors in their discussion, is the unclear relevance of these findings to humans. The focus is on a primary bile acid (TβMCA) that is unique to mice and mono-colonized mice are obviously an artificial system. The latter might be addressed by colonizing mice with a defined gut microbiota or introducing these genetically engineered strains to SPF animals. I think it's also important to discuss why human gut bacteria would have evolved enzymes that target mouse bile acids; perhaps consistent with some degree of enzyme promiscuity.

I also would have liked to see a more in-depth analysis of the identified *Bacteroides* BSH and the potential mechanism that determines its specificity. Are there any distinguishing characteristics of selective versus non-selective strains, either in terms of primary sequence, genetic neighborhood, or other factors? One way to get at this would be to express both types of genes in the same host to determine if this selectivity is intrinsic to the gene itself or depends on the background genetic context.

The screen presented in Figure 2 is a major part of the paper but could be presented in more detail and potentially multiple figures. It was not clear why 25% was used as the threshold instead of presenting the actual% conversion. No time curves are shown making it hard to evaluate if 48 hours represents steady state for these reactions. There are also no replicates shown making it impossible to determine the reproducibility of these results. It also seems important to test the full panel of compounds against the isogenic strains to enable a more definitive test of selectivity. This experiment should be done in replicate and include appropriate statistics (traces aren't sufficient).

The host phenotypic data was remarkable especially given the fact that only a single gene was altered (assuming no background mutations; see comment below). However, the mechanism of action gets pretty murky. The authors seem to rule out FXR, despite using it as a rationale for these experiments, raising the question of what the signaling pathway actually is. A couple of candidates are mentioned, but I'm skeptical as to the ability to definitively implicate these pathways based only on the transcriptional profiling data. Experiments in transgenic mice would go a long way towards testing the role of FXR or other nuclear receptors.

I was also concerned about some inconsistent results throughout the paper and the inclusion of a lot of non-significant "trends". In my opinion, one cannot assign any meaning to trends for these types of metabolic experiments, especially without any independent replication experiments. It also made the host phenotype sections really difficult to read and interpret due to the mixing of non-significant and significant results. It was also hard to keep track of which results were reproduced across both mouse experiments. Another major caveat is that *B. theta* was actually a poor metabolizer of TβMCA in Figure 2A, which is the focus of all the subsequent experiments.

Reviewer #2:

The manuscript by Yao et al., screened *Bacteroides* spp. for bile salt hydrolysis activity against a panel of conjugated bile salts and decided on *B. theta* which deconjugated TβMCA (FXR antagonist) but not TCA (FXR agonist). Since *B. theta* has a well-defined genetic system, the authors knocked out the gene encoding bile salt hydrolase (BSH). They then colonized germ-free C57BL/6 mice with WT vs. BSH KO vs. GF and determined that weight loss in the KO colonized mice was not due to differential colonization, nor differences in fecal caloric profile, nor differences in detergent properties of ileal bile acids between groups. This makes sense because it is likely that *B. theta* will colonize cecum, rather than ileum. Instead, the effects were due to signaling (FXR-dependent and FXR-independent) which may or may not be directly related to bile salt composition rather than say alterations in *B. theta* physiology upon loss of BSH activity. This is a well-executed study and well-written manuscript.

Reviewer #3:

In this report, Yao et al., characterize the contribution of microbiome bile salt hydrolase activity through studies of gnotobiotic mice mono-associated with wildtype *Bacteroides thetaiotaomicron* or an isogenic mutant lacking a candidate BSH gene. The authors report that these mice exhibit altered metabolism in many aspects, suggesting that the ability of gut microbes to carry out this activity could impact host biology.

1) The authors do not complement any of their mutations. It is unclear whether the genetic deletions cause phenotypes due to the absence of the targeted gene, polar effects on other genes, or random mutations that occurred elsewhere in the genome during the genetic engineering process. While it could be argued that such events are unlikely, complementation is an important part of microbial genetic analysis.

2) The authors do not demonstrate that the BT2086 protein has BSH activity.

3) The authors do not demonstrate that BT2086 is expressed in mice. Combined with the comments above, this makes it difficult to evaluate the conclusions of the paper.

4). In Figure 7B (global transcriptional analysis), the authors conclude that "samples form distinct clusters based on colonization status". It is unclear how the data proves this, as the samples do not show this clustering in the figure. Statistical support for this claim should be provided to demonstrate that the optimal number of clusters is two, and that these clusters separate samples based on colonization status.

---

## [Author Response]

Summary:All three reviewers were very enthusiastic about this manuscript and are in principle supportive of its publication pending revision in the upcoming eLife microbiome special issue. In this report, Yao et al. screened Bacteroides spp. for bile salt hydrolysis activity against a panel of conjugated bile salts and decided on *B. thetaiotaomicron* which deconjugated TβMCA (FXR antagonist) but not TCA (FXR agonist). Since *B. theta* has a well-defined genetic system, the authors knocked out the gene encoding bile salt hydrolase (BSH). Finally, they performed studies of gnotobiotic mice mono-associated with wildtype *B. theta* or an isogenic mutant lacking a candidate BSH gene. The two colonization groups exhibited altered metabolism in many aspects, suggesting that the ability of gut microbes to carry out this activity could impact host biology. This model is a great starting point for a more detailed dissection of the role of specific bacterial genes and metabolites in diet-induced obesity.

We’re so glad to hear that the reviewers were enthusiastic about this paper. We thank the reviewers for their time and careful reading of this manuscript and figures. Below please find the reviewers’ comments and concerns addressed in a point-by-point fashion.

Essential revisions:While there are many potential areas for future study, the reviewers agreed that one major issue that is necessary to address prior to publication is that the authors do not complement any of their mutations. It is unclear whether the genetic deletions cause phenotypes due to the absence of the targeted gene, polar effects on other genes, or random mutations that occurred elsewhere in the genome during the genetic engineering process. While it could be argued that such events are unlikely, complementation is an important part of any bacterial genetic analysis.

We agree with the reviewers that complementation to restore the wild-type profile of *Bacteroides thetaiotaomicron* would strengthen our finding that BT2086 is responsible for the bile salt hydrolase activity in this species. We did not originally perform this experiment as complementation is not standard practice for *B. theta*, perhaps due to the incomplete understanding of transcriptional and translation control in this organism (examples of papers where complementation was not performed for a *B. theta* knockout: Jacobson et al., 2018; Ng et al., 2013; Taketani et al., 2015.)

There are examples, however, in which *B. theta* has been successfully complemented (Cullen et al., 2015; Mishra and Imlay, 2013, Molec. Micriobiol.). Using the method of the Goodman and co-workers, we were able to successfully complement BT2086. We have added this result to Figure 2C and the figure legend: “Representative UPLC-MS traces showing that *Bacteroides thetaiotaomicron* wild-type (Bt WT) and BtΔ1259 deconjugate TUDCA whereas BtΔ2086 does not. BtΔ2086,2086^+^ recovered the deconjugation function while the BtΔ2086,CTRL^+^ control strain containing an empty pNBU2 vector did not, demonstrating that BT2086 is responsible for bile salt hydrolase activity in Bt.”

We have also made the following changes to the main text and Materials and methods section:

Changed subsection heading from “BT2086 is a BSH in *Bacteroides thetaiotaomicron*” to “BT2086 is responsible for BSH activity in *Bacteroides thetaiotaomicron”.*

Including the following in subsection “BT2086 is responsible for BSH activity in Bacteroides thetaiotaomicron”: “Complementation of the Bt KO strain with BT2086 restored BSH activity (Figure 2C), confirming that BT2086 is necessary for bile acid deconjugation in Bt.”

Including the following in subsection “Construction of *Bacteroides thetaiotaomicron* complementation strains”: “The *Bacteroides thetaiotaomicron* complementation strains were constructed using a previously described method with slight modifications. […] Recovery of function of the complementation strain was confirmed via UPLC-MS with 100 μM TUDCA as substrate.”

Additional caveats can likely be addressed in the Discussion section without the need for more experiments. This includes the unclear relevance of these findings to humans. The focus is on a primary bile acid (TβMCA) that is unique to mice and mono-colonized mice are obviously an artificial system. The authors should include more discussion about why human gut bacteria would have evolved enzymes that target mouse bile acids; perhaps consistent with some degree of enzyme promiscuity.

We appreciate the reviewers’ suggestion to expand and clarify the sections of the text discussing the question of human relevance. In our interpretation, it is not that human gut bacteria have evolved enzymes to target mouse bile acids. Previously known gut bacterial BSH hydrolyze all conjugated bile acids (regardless of host source). Rather, the *Bacteroides* species uncovered here display some selectivity and this means that they can cleave C12=H but not C12=OH primary bile acids. Mice and humans both possess bile acids within the “cleaved” and “not cleaved” groups.

We have added the following text to the Discussion section to clarify this point and also further expand on the human relevance of our findings:

“Since the majority of BSH characterized to date from Bacteroidetes and Firmicutes are promiscuous and do not display selective deconjugation activity based on the bile acid substrate, it is possible that selective BSH activity may be an evolved trait. The lack of distinct clustering of Group I (i.e., steroidal core-selective) BSH at both the strain and protein levels suggests this activity that may have arisen multiple times in evolutionary history from different bacterial hydrolase precursors. Structural comparisons of closely related BSH with different selectivity profiles may reveal individual amino acids that could be responsible for the activities observed.”

“While the major primary bile acids in mice are TβMCA and TCA (Sayin et al., 2013), humans produce glyco- and tauro-conjugated CDCA and CA (Russell, 2003). […] Our results suggest that *Bacteroides* species status in individuals may in part determine downstream bile acid pool composition in these people.”

The in vivo mechanism of action remains unclear, necessitating future studies. The authors should comment on the degree to which these phenotypes may be dependent or independent of FXR and discuss alternative mechanisms of action.

We have expanded the Discussion section to include a more detailed explanation of the FXR-dependence or FXR-independence of the phenotypes observed. We have also expanded the discussion of alternative mechanisms:

“The decreased expression of FXR target genes *Shp* and *Apoc2* as well as the increased expression of *Cyp7a1*, the rate-limiting enzyme in bile acid biosynthesis, are consistent with a regime of FXR antagonism in the livers of Bt KO- compared to Bt WT-colonized mice. […]

We cannot rule out the possibility that host receptors beyond FXR and PXR may be involved in the differences noted between Bt KO- and Bt WT-colonized mice.”

In terms of which bile acids may be affecting PXR pathways, that remains an open question that we plan to tackle in subsequent research. The secondary bile acid lithocholic acid (LCA) is a PXR agonist (Staudinger et al., 2001). As expected, we did not detect LCA in any mice in our studies, as none of these mice were colonized with bacteria known to produce this molecule (Ridlon et al., 2006). It is possible, then, that bile acids that were found at significantly different levels between Bt KO-colonized and Bt WT-colonized mice may play a role in PXR regulation. The further exploration of bile acids as either ligands for PXR or molecules that affect pathways that regulate PXR activation falls outside the scope of the current work and will be addressed in a future paper.

The authors should also include a more in-depth discussion of the identified Bacteroides BSH and the potential mechanism that determines its specificity. Are there any distinguishing characteristics of selective versus non-selective strains, either in terms of primary sequence, genetic neighborhood, or other factors?

In order to explore the question of the origin of selectivities observed in our 20-strain Bacteroidetes screen, we performed phylogenetic analyses at both the strain and protein levels. To help the reader follow our logic, we have now placed these 20 strains into 4 groups (now outlined in the Figure 2 legend): “Group I (red): Bacteroidetes species that deconjugate primary bile acids based on steroidal core structure (C12=H but not C12=OH); Group II (gray): species that deconjugate based on amino acid conjugate; Group III (blue): species that deconjugate all bile acid substrates; Group IV (black): no deconjugation observed.” We then analyzed the phylogenetic clusterings of these 4 groups. We added a new Figure (now Figure 3) describing the results. The lack of clustering observed limits further interpretations at this time. However, as we suggest in our Discussion section, these phylogenetic trees reveal groups or pairs of BSH that could be subject to future structural analyses.

The subsection “Bacteroidetes BSH exhibit evolutionary diversity” was added to Results section.

The following was added to the Discussion section: “Since the majority of BSH characterized to date from Bacteroidetes and Firmicutes are promiscuous and do not display selective deconjugation activity based on the bile acid substrate, it is possible that selective BSH activity may be an evolved trait. […] Structural comparisons of closely related BSH with different selectivity profiles may reveal individual amino acids that could be responsible for the activities observed.”

The subsection “Phylogenetic analysis of candidate BSHs and Bacteroidetes strains” was added to Materials and methods section.

The screen presented in Figure 2 is a major part of the paper but needs to be presented in more detail. It was not clear why 25% was used as the threshold instead of presenting the actual% conversion.

We initially presented the data in a crude heat map (no deconjugation, less than 25% deconjugation, more than 25% deconjugation) because our goal was to obtain a “yes/no” answer for whether a given Bacteroidetes strain deconjugated a given bile acid. Obtaining these “yes/no” answers for deconjugation abilities then allowed us to separate the 20 strains into groups based on their substrate preferences. We agree that adding more detail will make these heatmaps more useful to the reader. We now present the data as “full” heatmaps, with mean deconjugation values represented by colored shading (Figure 2A).

No time curves are shown making it hard to evaluate if 48 hours represents steady state for these reactions.

We agree that it is important to discuss why we decided to plot our heatmaps at a 48 hour timepoint. By taking timepoints from bacteria from the different activity groups, we ascertained that all metabolism had ceased by 48 hours. We have added a new panel (now 2B) to Figure 2 showing representative timecourses from a Group I strain (*B. theta*) and a Group III strain (*Bacteroides* sp. 1_1_6) to illustrate this finding. We have also added deconjugation heat maps from a 24 hour timepoint as a supplementary figure (Figure 2—figure supplement 3).

In addition, we made the following change to the main text (subsection “Selected species of *Bacteroides* accept distinct bile acid cores as BSH substrates”): “We monitored deconjugation over time by UPLC-MS and determined that all hydrolysis reactions had reached steady state by 48 hours (Figure 2B, Figure 2—figure supplement 3). We then quenched the cultures and profiled bacterial bile acid metabolism.”

There are also no replicates shown making it impossible to determine the reproducibility of these results. It also seems important to test the full panel of compounds against the isogenic strains to enable a more definitive test of selectivity. This experiment should be done in replicate and include appropriate statistics (traces aren't sufficient).

We included UPLC-MS traces in our initial submission because the use of representative traces is the standard way to represent data in the field of metabolite production by bacteria (examples: Yan et al., *eLife,* 2017. Figure 4B; Dodd et al., 2017. Figure 2D and H, Figure 3; Ganley et al., 2018. Figure 2.). We agree with the reviewers that it is important to show the reproducibility of these results. In our original Figure 2 legend, we specified that “Assays were performed in at least biological duplicate.” We have replaced the Figure 2 supplements showing representative UPLC traces with graphs showing the% conversion of these biological duplicates. As stated in above, the goal of these experiments was to obtain a “yes/no” deconjugation answer for each strain for a given substrate. There were no instances where in one experiment, a strain did not deconjugate a bile acid at all and in the second experiment, deconjugation was detected (or vice versa). As such, a “tie-breaker” third experiment was not necessary.

We tested the full panel of bile acid substrates against the isogenic strains (BtΔ2086 and BtΔ1259) and have added these results to the bottom of the heatmaps as well as Figure 2—figure supplements 1, 2, and 3. As expected, BtΔ2086 did not deconjugate any of the bile acid substrates and BtΔ1259 displayed Bt WT deconjugation abilities.

The authors should also address the major caveat that *B. theta* was actually a poor metabolizer of TβMCA in Figure 2A, which is the focus of all the subsequent experiments.

We have addressed this caveat and explained why we chose to focus on *B. theta* in the following text added to the Results section:

“All of the five Group I strains displayed weak to moderate deconjugation of TβMCA in vitro (Figure 2A).[…] Although this strain displayed relatively weak TβMCA-deconjugating activity, Bt had been previously shown to be amenable to genetic manipulation, allowing knockout of putative BSH genes (Cullen et al., 2015; Koropatkin, Martens, Gordon, and Smith, 2008).”

Finally, the manuscript needs a careful round of revisions. Multiple inconsistent results and non-significant "trends" can be trimmed down substantially. The host phenotype sections were particularly difficult to read and interpret due to the mixing of non-significant and significant results.

We have carefully revised the main text and corresponding figures in order to increase the clarity of this work and to cut down on the discussion of “trends.” We have also added labels to our figure panels to aid the reader in switching from the main text to the figures and back again. Major changes are as follows:

Figure 1: Highlighted C12, the key site of recognition for the selective Bacteroides BSH; wrote out “deoxycholic acid” and “lithocholic acid”.

Corresponding textual changes: Converted all references to R=H and R=OH to C12=H and C12=OH.

Figure 4 (was Figure 3): Added a schematic timeline for the monocolonization and CLAMS GF experiments (new A). Enlarged the bile acid bar graphs to make them easier to read. Moved liver and plasma bile acids to Figure 4—figure supplement 2.

Corresponding textual changes: Added references to Figure 4A to main text. Streamlined discussion of bile acid analyses and removed references to trends in cecal contents: “As we predicted, Bt KO-colonized mice displayed higher levels of TβMCA in cecal contents than Bt WT-colonized mice in the monocolonization experiment (Figure 4B). […] We observed the same significant difference in βMCA levels in feces (Figure 4C, red highlight boxes).”

We are still including our discussion of trends in the distal ileum because we feel that we explain how biological differences may explain the variation in bile acid values observed.

Figure 5 (was Figure 4): In Figure 5B, converted CFU/g from a linear to a log scale and removed the GF condition from these plots. Added the words “plasma” and “liver” to Figure 5C and D, respectively, for a quick visual reference of tissue type for the lipid analyses.

Figure 6 (was Figure 5): Moved the locomotor activity panels to Figure 6—figure supplement 1. Switched the CO_2_ production and O_2_ consumption panels to correspond to the reworded main text.

Corresponding textual changes: Removed the discussion of trends in locomotor activity, clarified the CO_2_ / O_2_ discussion as related to RER:

(Subsection “Bt BSH status affects host metabolic indications”) “While Bt KO-colonized mice consumed more oxygen than Bt WT-colonized mice (Figure 6B), there were no significant differences in carbon dioxide production between groups (full day, Bt WT vs Bt KO p=0.4041; Bt WT vs GF p=0.3239; Bt KO vs GF p=0.0606) (Figure 6C). These data are consistent with the lower RER observed in Bt KO-colonized mice. No statistically significant differences in locomotor activity were noted between the three groups (Figure 6—figure supplement 1).”

Figure 8 (was Figure 7): Enlarged the pictures and text in panels Figure 8A and B. Moved plasma glucose to new Figure 9 and rearranged Figure 8C and D (now panels C, D, and E). Split panel D into two panels (now D and E) based on the biological functions of the genes. Added labels to panels C (Distal ileum, RNA-Seq), D (Distal ileum, glucose and lipid metabolism gene expression), and E (Distal ileum, circadian rhythm gene expression). Removed results that were not statistically significant (i.e., *Slc2a1, Cry1, Cry2* gene expression).

Corresponding textual changes: Removed the reference to qPCR of *Slc2a1*.

Figure 9 (was Figure 8): Moved glucose, insulin, leptin, and ghrelin to panel A and added labels above each plot. Removed liver expression of FXR and circadian rhythm genes (most not statistically significant). Grouped liver genes into FXR-mediated and FXR-independent groups. Added not significant (ns) bars where relevant to clarify statistical outcomes. Added a p value to liver *Srebp2*. Added labels to panels B and C (Distal ileum, FXR-mediated gene expression; Liver, FXR-mediated gene expression and Liver, FXR-independent gene expression).

Corresponding textual changes: Changed the concluding sentence of the first paragraph in the Results section (subsection “Bile acid pools alter the expression of FXR-dependent and FXR-independent genes in the liver and distal ilem”): “Taken together, our data are consistent with a scenario in which bile acid-mediated FXR antagonism is affecting pathways in the liver but not the ileum of BT KO-colonized mice.”

Clarified and shortened the paragraph discussing FXR-independent signaling in the Results (subsection “Bile acid pools alter the expression of FXR-dependent and FXR-independent genes in the liver and distal ileum”): “In addition, we observed significant decreases in genes not known to be under the control of FXR, including *Cd36* (p=0.0015), a gene encoding a fatty acid transporter, the immune-related genes tumor necrosis factor α (*Tnfα*, p=0.0225) and EGF-like module-containing mucin-like hormone receptor-like 1 (*Emr1*, p=0.0011), and the G-protein coupled receptor S1pr2 target gene sphingosine kinase 2 (*Sphk2*, p=0.0274) (Nagahashi et al., 2015), in the liver of Bt KO-colonized mice (Figure 9C). These results indicate that other host receptors may be involved in the transcriptional changes and metabolic differences observed.”

It was also hard to keep track of which results were reproduced across both mouse experiments, which could be addressed by a supplemental table or summary diagram.

(See above): We added a schematic timeline for the monocolonization and CLAMS GF experiments (new Figure 4A).

We also included a new table (Figure 4—table supplement 1) outlining which analyses were performed using which mouse experiment(s).

Reviewer 1 had also raised this point: *[…] The host phenotypic data was remarkable especially given the fact that only a single gene was altered (assuming no background mutations; see comment below). However, the mechanism of action gets pretty murky. The authors seem to rule out FXR, despite using it as a rationale for these experiments, raising the question of what the signaling pathway actually is. A couple candidates are mentioned, but I'm skeptical as to the ability to definitively implicate these pathways based only on the transcriptional profiling data. Experiments in transgenic mice would go a long way towards testing the role of FXR or other nuclear receptors.*

We considered performing an experiment in germ-free FXR knockout mice but decided against it for two main reasons (one biological, the other technical). First, research has shown that on both normal chow and a high-fat diet, FXR KO mice gain significantly less weight than wild-type mice (Prawitt et al., 2011, Figure 5A). HFD-fed FXR KO mice also display lower glucose and insulin levels. The difference in phenotype of FXR KO mice monocolonized by Bt KO compared to Bt WT would likely be masked by this strong metabolic FXR KO phenotype, therefore eliminating the usefulness of performing this experiment. Second, there would be considerable cost, time, and expertise required to re-derive FXR KO mice germ-free and also to maintain a breeding colony of these FXR KO mice once they were GF. Based on these two factors, we decided against performing an experiment in GF FXR KO mice.